# Extensive gene content variation in the *Brachypodium distachyon* pan-genome correlates with population structure

Sean P. Gordon[1], Bruno Contreras-Moreira [2,3,4], Daniel P. Woods[5,6], David L. Des Marais [7,17], Diane Burgess[8], Shengqiang Shu[1], Christoph Stritt[9], Anne C. Roulin[9], Wendy Schackwitz[1], Ludmila Tyler[10], Joel Martin [1], Anna Lipzen[1], Niklas Dochy [11], Jeremy Phillips[1], Kerrie Barry[1], Koen Geuten [11], Hikmet Budak [12], Thomas E. Juenger[13], Richard Amasino [5,6], Ana L. Caicedo [10], David Goodstein [1], Patrick Davidson[1], Luis A. J. Mur [14], Melania Figueroa [15], Michael Freeling[8], Pilar Catalan [4,16] & John P. Vogel [1,8]

While prokaryotic pan-genomes have been shown to contain many more genes than any individual organism, the prevalence and functional significance of differentially present genes in eukaryotes remains poorly understood. Whole-genome de novo assembly and annotation of 54 lines of the grass *Brachypodium distachyon* yield a pan-genome containing nearly twice the number of genes found in any individual genome. Genes present in all lines are enriched for essential biological functions, while genes present in only some lines are enriched for conditionally beneficial functions (e.g., defense and development), display faster evolutionary rates, lie closer to transposable elements and are less likely to be syntenic with orthologous genes in other grasses. Our data suggest that differentially present genes contribute substantially to phenotypic variation within a eukaryote species, these genes have a major influence in population genetics, and transposable elements play a key role in pan-genome evolution.

[1] DOE Joint Genome Institute, Walnut Creek, CA 94598, USA. [2] Estación Experimental de Aula Dei-CSIC, 50059 Zaragoza, Spain. [3] Fundación ARAID, 50018 Zaragoza, Spain. [4] Grupo de Bioquímica, Biofísica y Biología Computacional (BIFI, UNIZAR), Unidad Asociada al CSIC, 500018 Zaragoza, Spain. [5] University of Wisconsin, Madison, WI 53706, USA. [6] United States Department of Energy Great Lakes Bioenergy Research Center, Madison, WI 53726, USA. [7] Harvard University, Cambridge, MA 02138, USA. [8] University of California, Berkeley, Berkeley, CA 94720, USA. [9] University of Zürich, Zürich, CH-8006, Switzerland. [10] University of Massachusetts Amherst, Institute for Applied Life Sciences, Amherst, MA 01003, USA. [11] University of Leuven, KU Leuven, Leuven, 3000, Belgium. [12] Montana State University, Bozeman, MT 59717, USA. [13] University of Texas Austin, Austin, TX 78705, USA. [14] Aberystwyth University, Aberystwyth, SY23 3FL, UK. [15] University of Minnesota, St. Paul, MN 55108, USA. [16] Universidad de Zaragoza-Escuela Politécnica Superior de Huesca, 22071 Huesca, Spain. [17] Present address: Department of Civil and Environmental Engineering, Massachusetts Institute of Technology, Cambridge, MA 02139, USA. Correspondence and requests for materials should be addressed to J.P.V. (email: jpvogel@lbl.gov)

The genetic variation found among individual members of a species is the raw material upon which natural and artificial selection act. This variation ranges from single-nucleotide polymorphisms (SNPs) to large presence/absence variants (PAVs) encompassing multiple genes. Most genome-wide studies of intra-specific genetic variation in eukaryotes use reference-based approaches in which individual sequence reads from different accessions are compared to a single reference genome. This approach efficiently identifies SNPs and small insertions/deletions (indels) in reasonably well-conserved regions as well as larger deletions. In contrast, long DNA sequences that are not found in, or are highly diverged from, the reference genome are overlooked by most analyses. Although the amount, biological and physical properties, as well as turnover of such sequence is poorly understood in eukaryotes, several studies indicate that may be responsible for phenotypes of high value to breeders and biologists[1–4]. Indeed, in rice 41.6% of trait-associated SNPs from genotype-by-sequencing markers were not contained in the reference genome[5]. Unlocking the information in non-reference sequences necessitates defining the extant catalog of genetic diversity within a species, termed a pan-genome[6].

Pan-genomes have been created for some bacterial species and, typically, are much larger than the genome of any individual strain[6]. In contrast, the challenges associated with creating multiple high-quality eukaryotic de novo genome assemblies and associated sequence annotations have prevented large-scale, in-depth exploration of eukaryotic pan-genomes. Rather, eukaryotic pan-genome studies have employed several approaches to avoid the difficulty of generating many high-quality genome assemblies: using reference genome-based approaches with targeted de novo assembly[7–12], focusing on a small number of relatively low-quality de novo assemblies[13,14]; employing a metagenome approach that combines low depth sequences from many lines with targeted de novo assembly[5]; or creating a pan-transcriptome as a way to reduce complexity[15–17]. While these studies all have limited ability to capture and describe the full nuclear pan-genome, most suggest a pan-genome that is considerably larger than the genome of any individual line. For example, a study of the maize pan-transcriptome suggested that the reference genome only contained half the genes in the maize pan-genome[15], a study of the low-copy regions of 18 wheat lines found 21,653 predicted genes that were not contained in the reference genome despite the fact that the lines were closely related[9], and a metagenome study of rice found 8,000 genes that were not in the Nipponbare reference genome[5]. The rice study also performed a genome-wide association study that showed a remarkable 41.6 % of trait-associated SNPs were from genomic locations corresponding to non-reference genes. Thus, plant pan-genomes are potentially large and a source of important traits.

To enable detailed insights into the size, phylogenetic distribution, importance and evolution of eukaryotic pan-genomes, we created a pan-genome for the model grass *Brachypodium distachyon*, whose small (272 Mbp), diploid, highly homozygous genome simplifies genomic studies[18,19]. Further enabling our study, the *B. distachyon* reference genome is a finished genome except for the placement of some centromeric repeats[20]; a large collection of publicly available, geographically, genetically and phenotypically diverse natural accessions has been assembled[18,21]. Since *B. distachyon* is not domesticated, its pan-genome represents an unaltered view of intra-species natural diversity. Finally, since this small, wild, annual grass has been widely adopted as a model system for the grasses used to produce grains, forage, and biofuels, numerous experimental resources exist to functionally test hypotheses developed from the analysis of its pan-genome[19,22,23].

## Results

**De novo genome assembly and pan-genome construction.** We sequenced (92x median genome coverage, 100 bp paired-end Illumina short reads) and assembled the genomes of 54 diverse *B. distachyon* inbred lines (Supplementary Table 1)[18,19,21]. As a control, we assembled Illumina sequence data from the same line, Bd21, used to create the reference genome[22]. The mean assembled genome size was 268 Mbp, very close to the 272 Mbp reference genome size[22] (Fig. 1a, Supplementary Table 2). We achieved a scaffold L50 of 1 Mbp for the best assembly and an average of 75 kb. Our best de novo assembly has a higher percent completeness[24,25] (Bd18-1: BUSCO score 98.4%) than the v2.0 *B. distachyon* reference genome sequence (98.3%), which is among the top ten most complete plant reference genomes (Supplementary Table 2, Supplementary Fig. 1a). Although five assemblies were less complete than average, mainly in non-coding sequence (Fig. 1a), all assemblies were more complete than some recently published reference genomes and contained similar levels of PAV (Supplementary Fig. 1a, Supplementary Table 2). Scaffolds were ordered and oriented into five chromosome-scale assemblies using synteny to the reference genome for each line. Up to 97% (72% on average) of the assembled bases for each line was placed into chromosomes[26] (Fig. 1b). Characterization of our assembled sequence using flanking genes and synteny enabled us to distinguish allelic and isoform differences from gene content differences, which distinguishes this work from pan-genomes based on transcriptome data or a metagenome.

To validate PAVs, we mapped raw reads onto various assemblies. Despite being supported by their own raw reads, a median of 1 Mbp of genic sequence and greater than 8 Mbp of non-genic sequence (~ 3% of each genome) from each assembly lacked reference control read alignments (Fig. 1c, Supplementary Fig. 1c, Supplementary Note 1). An example of a region encompassing ~ 15 kb of non-reference sequence that contains an annotated gene is shown graphically in Fig. 1d. Read alignments to this region clearly demonstrate that it is found in some lines but not others (Fig. 1e). We also observed sequences spanning hundreds of kilobases that are absent or extremely diverged from the reference genome (Fig. 1f). Alignment of short reads from each line to the reference genome indicated a similar magnitude of sequence absence (Fig. 1c, Supplementary Fig. 1)[27].

To obtain a preliminary estimate of pan-genome size, purely at the DNA sequence level, we constructed a sequence-based pan-genome by iteratively comparing each of the 54 genome assemblies to the preceding pan-genome to identify novel sequences >600 bp (long enough to contain a gene). We defined sequence as novel if it did not contain a single 21 bp sequence found in the preceding pan-genome (Supplementary Note 1). The sequence-based pan-genome was 430 Mb, 58% larger than the 272 Mb reference genome, and contained 40% more genes. The average length of the DNA segments added to the pan-genome was 1,487 bp, much larger than the 600 bp minimum length cutoff. These analyses reveal a large amount of gene and non-coding sequence that is not captured by a single reference genome.

To enable a more detailed and contextual biological analysis, a pan-genome based on annotated genes was also constructed. First, each assembly was individually annotated by the same pipeline used to annotate the reference genome. The number of genes and total length of coding sequence identified for each line were similar to the reference annotation, further indicating that the assemblies and annotations are essentially complete (Fig. 1a, Supplementary Table 2). As a control for the assembly and annotation processes, we annotated our Bd21 short read assembly and the Sanger-based Bd21 reference assembly (v2.0), resulting in three annotations for Bd21: the v2.1 reference annotation from

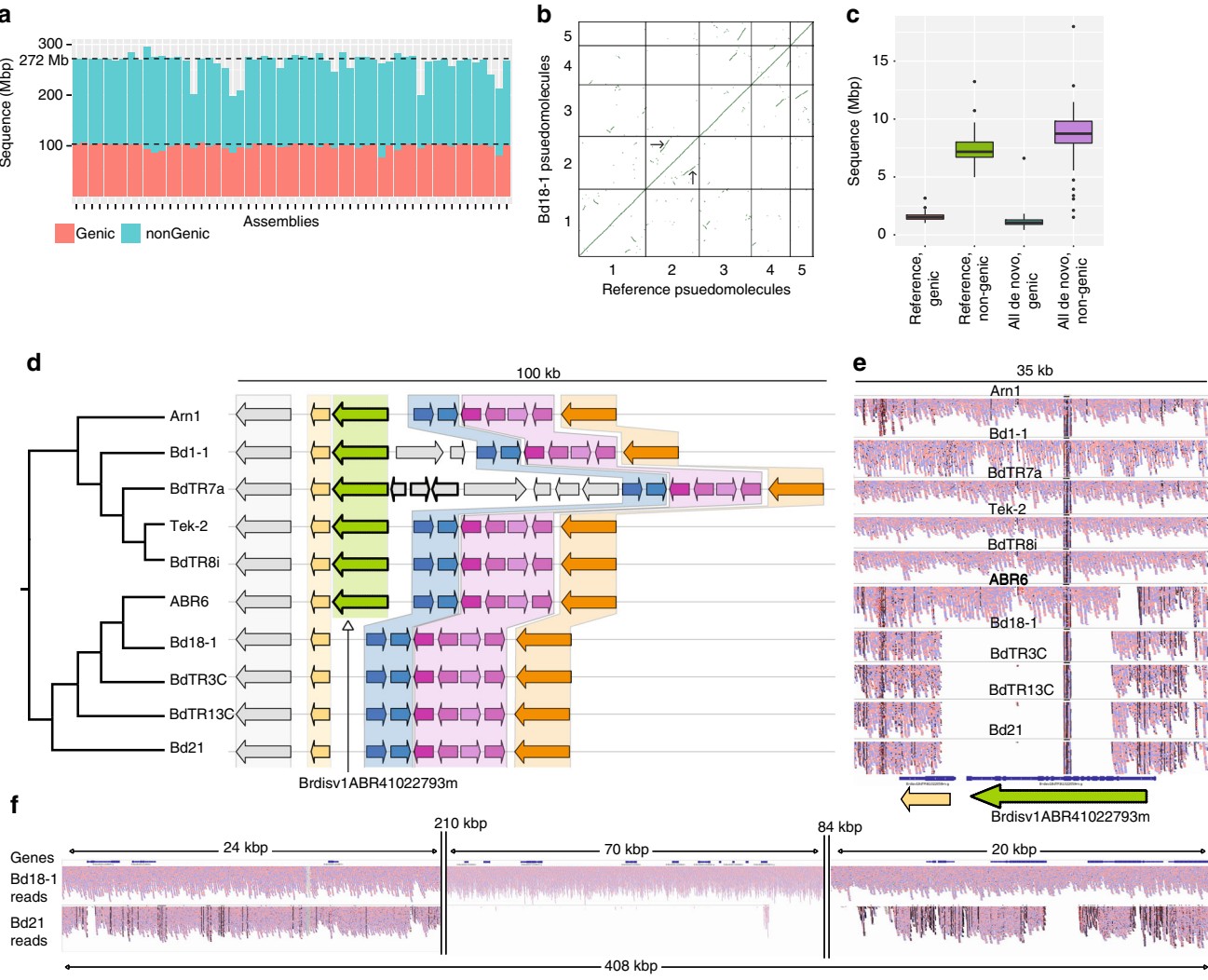

**Fig. 1** Genome assembly and analysis. **a** Genome assembly size and coding sequence for all lines. Dashed lines correspond to the reference genome size and amount of coding (red) and non-coding (blue) sequence. See Supplementary Table 2 for line names and assembly statistics. **b** Dotplot of syntenic genes between Bd18-1 scaffolds and the reference genome. Note that the short syntenic segments (one example indicated by arrows) off the main diagonal line are signatures of an ancient whole-genome duplication. These short segments are apparent even when comparing the reference genome to itself[22] and are not assembly artifacts. **c** Similar amounts of coding and non-coding sequence are estimated to be absent in each line, measured by coverage of short reads from respective lines to the reference genome sequence, or alignment of reference short reads to respective de novo assemblies. Whiskers show data within 1.5 times the interquartile range (IQR). **d** Syntenic representation of the locus contining the non-reference pan-gene *BrdisvABR41022793m*. Note that the pan-gene is present in the top six genomes but absent from the bottom four, including the reference genome (Bd21). **e** Short reads from the lines in **d** mapped to the BdTR8i genome. Note that read mapping supports the presence/absence of *BrdisvABR41022793m*. **f** Short reads from Bd21 and Bd18-1 mapped to the Bd18-1 genome in a region where Bd18-1 contains 408 kb of sequence that is extremely diverged or absent from the reference genome (between the relatively conserved left and right flanking regions). Note that the interval contains multiple annotated genes

Phytozome[20], the re-annotated Bd21 v2.0 assembly (Bd21 annotation control), and an annotation of the Bd21 short read assembly created in this study (Bd21 assembly control). Overall, these controls were in agreement in all of our analyses, indicating that our assembly and annotation were accurate. Since genes with repetitive elements are the most problematic in short read assemblies, we examined one class of repetitive genes, those encoding nucleotide-binding-site leucine-rich-repeat proteins (NBS-LLRs), in detail. All annotations (including the reference annotation) contained similar numbers of annotations related to these genes (Supplementary Fig. 1f,g,e). Ninety-one percent of known reference NBS-LRR genes[22] had identical copy number between the assembly control and the reference, indicating that our approach was highly accurate (Supplementary Fig. 1e). As expected, the majority of observed differences were in multi-copy

NBS-LRR genes. Only a single (1/119) false PAV was detected in total and belonged to a single-copy reference NBS-LRR gene, which was not detected in the assembly control. Thus, our assembly and annotation approach was highly accurate and the most common problem, collapsed gene clusters, would actually decrease our estimates of pan-genome size.

Gene models were grouped based on sequence similarity using Markov clustering in the GET_HOMOLOGUES-EST pipeline[17,28]. This resulted in 61,155 pan-genome clusters that we categorized based on the number of lines in each cluster[29]: core gene clusters contained all lines, including the two reference controls (56 annotations total); softcore gene clusters contained 53–55 lines (95–98%); shell gene clusters contained 3–52 lines (5–94%); and cloud gene clusters contained 1 or 2 lines (2–5%) (Fig. 2a). This classification system allows us to represent some of

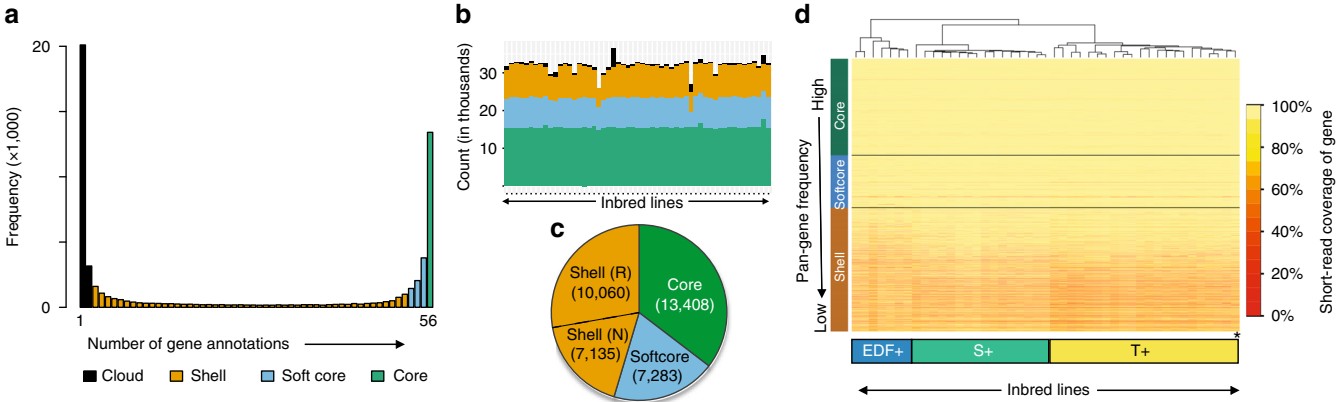

**Fig. 2** Gene-based pan-genome. Annotated genes (genomic sequence) from all genomes were clustered, and a single representative from each cluster was selected to create a gene-based pan-genome. **a** Number of pan-gene clusters represented within respective numbers of inbred line annotations. **b** Number of core, softcore, shell, and cloud pan-genes for individual inbred lines. **c** Number of core, softcore, and shell pan-genes in the high-confidence pan-genome. Shell pan-genes are divided into reference (R) and non-reference (N). **d** Percent coverage of 37,886 high-confidence pan-genes by short read data sets from 49 lines. Color-coded bars on the lower axis indicate the population groups described in Fig. 4; pan-genome categories are labeled on the vertical axis. Note that short read coverage supports the classification of the pan-genome compartments and that the clustering of lines by short read coverage matches the population groups identified in Fig. 4

the technical uncertainty, particularly for the softcore, where true core genes could be placed if they were missed in a few annotations, and the cloud, which could contain anomalous annotations from one or two genomes. Thus, the most robust comparison is between shell genes and core genes. We selected one gene from each gene cluster (termed a pan-gene) to make the gene-based pan-genome that was analyzed further. Since cloud genes are more likely to be assembly or annotation artifacts, they were excluded from subsequent analyses that focused on a high-confidence pan-genome consisting of 37,886 core, softcore, and shell pan-genes (Fig. 2b). Nevertheless, RNA-Seq data from the leaves of 36 lines indicated that 22% of cloud pan-genes were expressed, suggesting that a large number of cloud genes are bona fide genes. On average, any individual line in our study is composed of mostly core or softcore pan-genes (73%) and only 27% is categorized as shell or cloud (Fig. 2b). Thus, the majority of genes within any individual are found in all (or almost all) other individuals. Figure 2b also indicates that only one line, BdTR11a, contributes disproportionately to the number of cloud genes; this contribution does not appear to be explained by poor assembly quality.

Forty-five percent of the high-confidence pan-genes were shell genes, indicating that a sizable fraction of the genetic diversity in *B. distachyon* is not accessible via a reference-based strategy. Forty-one percent (7,135/17,195) of shell pan-genes are not contained in the v2.1 reference annotation, the v2.0 reference re-annotation control or the annotation of the short read assembly control (Fig. 2c). Forty-two percent of non-reference pan-genes had BLAST matches (E-value ≤ 0.0001) to plant species, and 78% of those hits were to plants other than *B. distachyon*. The largest number of non-*B. distachyon* hits were to the closely related grass, *Aegilops tauschii*, one of the three progenitors of bread wheat (Supplementary Fig. 2a). Less than 0.3% percent of non-reference sequences had matches to possible microbial sequences, ruling out contamination as a significant error.

We cross-validated the classification of core, softcore, and shell pan-genes by mapping raw reads from each assembly onto the pan-genes. As expected, raw read coverage of the pan-genes supports the presence/absence variation in shell and softcore pan-genes (Fig. 2d). Furthermore, clustering lines by short read coverage of respective pan-genes correctly identifies the three major population groups in this study, described in more detail

below. Thus, the observed PAVs are not simply due to assembly or annotation errors. To examine how many lines need to be sampled to capture the *B. distachyon* pan-genome, we conducted simulations of pan-genome size from increasing numbers of randomly selected lines. The results indicate that the pan-genome increases rapidly up to 20 lines and is still steadily increasing at 54 lines (Supplementary Fig. 2b). Thus, even sampling of 54 lines, chosen to represent the available genetic diversity, does not capture the full pan-genome of the species.

To gain insight into the evolution and estimate the false discovery rate of non-reference shell genes, we attempted to force gene models (lift-over) for non-reference pan-genes onto the v2.0 reference genome assembly, requiring ≥70% amino acid similarity between original and lifted-over peptides. Forty-two percent of non-reference pan-gene models (2988) could not be lifted over to the reference genome sequence. Fifty-two percent (1549/2988) of these could not be lifted over because there was no similar sequence in the reference genome. Forty-eight percent (1439/2988) of non-reference pan-genes that were not lifted over appear to be pseudogenes in the reference line identified by either partial peptide alignments (807) or incomplete lift-over models (632). Six percent (407) of lifted-over gene models resulted in peptides with less than 80% identity to the pan-gene model and therefore may be functionally distinct even if they are bona fide genes in the reference genome. Since we cannot be sure if the remaining lifted-over genes (3740) are genes in the reference genome without additional data, this represents the upper bound of our false positive rate for non-reference shell genes. However, given that the vast majority of shell pan-genes were confirmed as missing from two or more assemblies by read mapping, our overall estimate of the number of shell pan-genes is robust despite the fact that some genes may be missed in some annotations (Fig. 2c).

**Functional characterization of the pan-genome.** A comparison of the predicted biological functions of core and shell pan-genes revealed that core pan-genes are enriched for essential cellular processes (e.g., glycolysis), whereas shell and softcore pan-genes are enriched for functions that may be advantageous in some environments (e.g., gene regulation, disease resistance) (Fig. 3a, b). Genes such as Type I MADS-box and F-box genes that are known from inter-specific and subgenome fractionation comparisons to

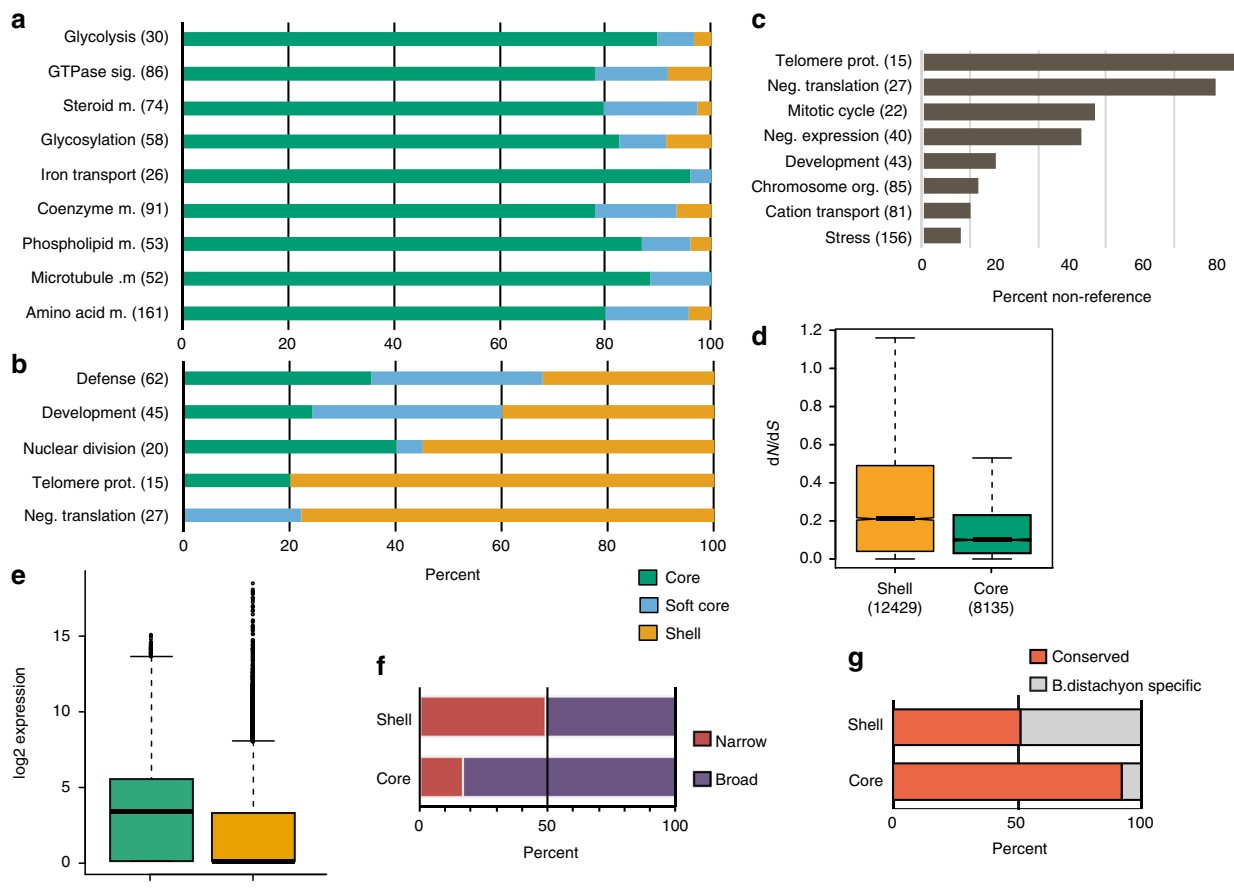

**Fig. 3** Functional classification of pan-gene categories. Gene Ontology (GO) biological process categories enriched in the core **a** and shell **b** pan-gene subsets, showing the distribution of all genes in that category across the pan-genome. Note that the "neg. (negative) translation" category includes defensive peptides that inhibit translation. **c** Percent of respective GO biological process categories comprised of non-reference genes. Total number of genes in each category is listed after the category label on the y axis. **d** The ratio of non-synonymous to synonymous mutations indicates that shell genes are evolving faster than core genes within *B. distachyon* ($p < 2.2e-16$, t-test). **e** Core genes are expressed at higher levels than shell genes ($p < 2.2e-16$, Wilcoxon signed rank). **f** Core genes are more broadly expressed within multiple tissues than shell genes and **g** are more likely to be identified as conserved in rice or sorghum. Whiskers in the above plots extend to the most extreme data point which is no more than 1.5 times the IQR

undergo accelerated birth, death and evolution[30] were also found at higher frequency among the shell pan-genes compared to core pan-genes, indicating that these genes are also more dynamic within a single species. Gene Ontology (GO) term enrichments for shell pan-genes include "negative regulation of translation" ($p < 0.05$, Fisher's exact test; false discovery rate (FDR) < 0.05), encoding rRNA N-glycosylase ribosome-inactivating proteins (RIPs). Such genes may be adaptive given that they are known to have a specific function in protection against attack from pathogenic fungi or herbivorous insects[31]. Expanded RIP gene families with unusually diverse domain architecture are common in Poaceae/cereal species[32]. GO enrichments were also observed for "telomere maintenance", associated with telomeric proteins known to show rapid evolutionary divergence and poor inter-species conservation, although this enrichment did not pass our FDR threshold ($p < 0.05$, Fisher's exact test; FDR > 0.05). Previous studies have shown that gene duplication has created telomere protein paralogs with novel functions[33].

As expected, non-reference pan-genes fall almost exclusively within the shell gene subset and have predicted functions similar to shell pan-genes (Fig. 3c). Shell pan-genes have higher non-synonymous/synonymous substitution ratios than core genes ($p < 2.2e-16$, Welch two-sample t-test, Fig. 3d) and an overall higher frequency of non-synonymous and synonymous

substitutions (Supplementary Fig. 3) suggesting reduced functional constraint. Compared to shell genes, core genes are generally expressed at higher levels ($p < 2.2e-16$, Wilcoxon signed rank test), and are more broadly expressed across tissues (Fig. 3e, f). Shell genes are less likely to have orthologs in rice and sorghum (Fig. 3g). All of this is consistent with the core gene set being enriched for genes that are under purifying selection because they perform essential functions and shell genes being less evolutionarily constrained because they are not essential under at least some environmental conditions. The observed enrichment of shell genes with putative adaptive functions suggests that shell genes are preferentially retained when they acquire functions that confer benefits under some conditions. Thus, shell genes may be responsible for considerable phenotypic variation that could be of particular interest for breeding improved crop varieties and evolutionary studies of adaptive traits.

**Population analysis**. To provide a phylogenetic benchmark for analysis of the pan-genome, we reconstructed a maximum likelihood (ML) tree based on 3,933,264 high-confidence SNPs (Fig. 4a, Supplementary Fig. 4a) that was consistent with previous results based on simple-sequence-repeat and genotyping-by-

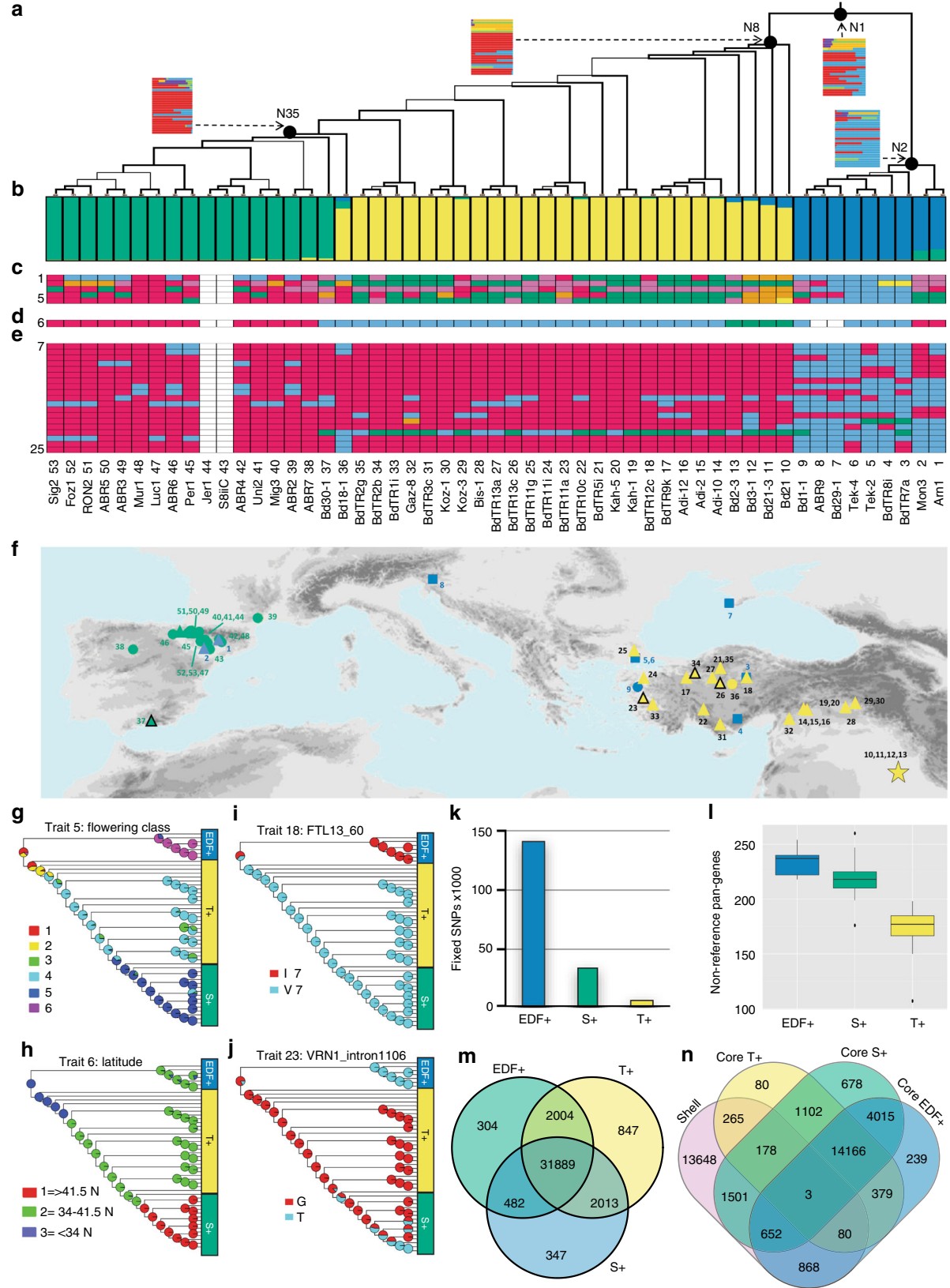

sequencing markers)[18,19,34,35]. To investigate the possibility that incomplete lineage sorting and the use of a single concatenated SNP data set[36] introduced topological errors in the phylogeny, we also ran a coalescence modeling phylogenetic analysis based on the Singular Value Decomposition quartets (SVDq)[37] method.

This approach is especially useful in cases with ongoing gene flow across the species range. Since the topology of the SVDq tree was very similar to the ML tree (Supplementary Fig. 4a,b), we focused on the ML tree for further analysis. The ML tree splits into two highly supported clades, one of which can be further separated

into two population groups (Fig. 4a, b). The first clade contains lines from multiple geographic locations that almost all exhibit an extremely delayed flowering (EDF+) phenotype (Fig. 4c). In contrast, the second clade contains groups corresponding to a set of eastern (predominantly Turkish, T+) lines and a set of western (predominantly Spanish, S+) lines that do not show the EDF+ phenotype (Fig. 4c, Supplementary Fig. 4a). The T+ and S+ groups come from mixed latitudes (Fig. 4d), and are differentiated in genotypes containing molecular variants within genes known to regulate flowering relative to those shown by individuals of EDF+ clade (Fig. 4e). The strongly supported differentiation of the S+ and T+ groups indicates that geographical isolation is another important factor in the divergence of the populations (Fig. 4f). We examined genetic structure and found that three population groups matched the ML tree (Fig. 4b) and that there was evidence of significant admixture in some lines (Supplementary Table 3).

To explore the evolutionary history of the population groups, we reconstructed the ancestral state for 25 discrete flowering time traits (Fig. 4c) and molecular variants in genes of known flowering regulators recorded in 53 of the sequenced lines (Fig. 4e, g, i, j Supplementary Figs. 5 and 6a). Changes were inferred to have occurred for most analyzed traits along the long branch leading from the most recent common ancestor (MRCA) of *B. distachyon* (node 1) to the MRCA of the EDF+ clade (node 2), which showed an ancestral pattern, congruent with that observed today in most of its descendant lines (e.g., extremely delayed flowering and distinct polymorphisms in vernalization and flowering genes; Fig. 4a, g, i, j, Supplementary Figs. 5, 6a, Supplementary Table 4). By contrast, changes were less pronounced in the short branches leading to the respective MRCAs of the T+ -S+ clade (node 8) and the S+ subclade (node 35) (Fig. 4a, g, i, j, Supplementary Figs. 5, 6a; Supplementary Table 4). In addition, Bayes factor (BF) tests revealed strong evidence (BF > 5) of correlation between flowering time traits (Supplementary Fig. 6b) and molecular variants in genes of known flowering regulators (Fig. 4e, i, j) (Supplementary Fig. 5, Supplementary Table 4,5, Supplementary Note 2). These results support the idea that flowering time is a major factor in the divergence of populations[10,27] (Fig. 4a–c, e, g–j Supplementary Fig. 5, 6a; Supplementary Table 5–7). Conversely, there was weak correlation (BF < 2) between collection latitude and flowering phenotype and polymorphisms in flowering and vernalization genes (Fig. 4d, h, Supplementary Fig. 5, 6b), indicating that latitude is not a major factor driving intra-specific divergence in *B. distachyon*.

$F_{ST}$ estimates were 0.5579, 0.6277, and 0.4220 respectively for EDF+|T+, EDF+|S+, and S+|T+ comparisons, indicating that the groups are strongly differentiated. Fixed-SNP differences were greatest between the EDF+ group and the other two groups (Fig. 4k). As expected, the EDF+ clade contributed the most non-reference gene clusters on a per-line basis due to the reference genome residing in the distant T+ group (Fig. 4l). We observed hundreds of genes that were present in only one of the population groups (Fig. 4m). In order to further explore the interplay between population structure and the pan-genome, we constructed pan-genomes using only non-admixed lines within the three population groups. As expected from our simulations (Supplementary Fig. 2a), the size of the population group pan-genomes decreased with sample size: T+=48,481 pan-genes among 21 lines; S+=38,739 pan-genes among 16 lines; EDF +=37,742 pan-genes among 7 lines. However, the number of non-reference genes in each of the three experiments was still significant. For the EDF+ group, we identified 2,868 non-reference pan-genes (relative to the EDF+ reference line BdTR8i); in the T+ group, we identified 6,746 non-reference pan-genes (relative to the non-admixed T+ reference BdTR3c); and in the S + group, we identified 4,790 non-reference pan-genes (relative to the S+ reference ABR4). In addition, we observed the same pattern as for the full pan-genome when we plotted the number of genes vs. the number of lines containing each pan-gene (Fig. 2a, Supplementary Fig. 2c). This indicates that the ratio of the various pan-gene compartments is not determined by population structure. Interestingly, we found that some shell genes in the full pan-genome are core within the sub-populations: 1,603 pan-genes for the EDF+ population; 2,334 pan-genes for the S+ population; and 526 pan-genes for the T+ population (Fig. 4n). Shell pan-genes that were core to the EDF+ population were enriched in GO terms including "regulation of gene expression" ($p < 0.05$, Fisher's exact test; FDR < 0.05), which was also observed for shell pan-genes core to the S+ population ($p < 0.05$, Fisher's exact test; FDR > 0.05). Shell genes core to either the T+ population or S+ population had enrichment for "multicellular organism development" ($p < 0.05$, Fisher's exact test; FDR > 0.05). Despite strong population structure, 6 of 53 lines (11%) in our study show a significant degree of admixture (Supplementary Table 3), including large stretches of DNA in the earliest flowering line Bd21 (also the source of the reference genome) that look very similar to EDF+ lines, indicating gene flow between the latest and earliest flowering lines. Not surprisingly, then, all three groups are fully inter-fertile in laboratory crosses. Thus, despite the high $F_{ST}$ rates and genomic structure, this group still appears to functionally behave as a single species.

Underscoring the accuracy of the pan-genome and our genome assemblies, dendrograms created based on PAVs, copy number variation, and average nucleotide identity recovered the same main population groups (EDF+, T+ and S+) identified by the

**Fig. 4** Populations analysis. **a** Maximum likelihood phylogenetic tree based on 3,933,264 SNPs for 53 *B. distachyon* lines. Thickness of branches indicates bootstrap support (thick, 100%; intermediate, 70–99%; thin, 50–69%). Insets at select nodes (N) show the probabilities for the ancestral state of the traits in **c** and **e**. **b** Plot of individual membership (SNP profiles) to optimal K = 3 Bayesian STRUCTURE groups: EDF+ (blue), T+ (yellow), S+ (green) (see Supplementary Table 3). **c–e** Color-coded matrix based on mapping all trait values to discrete state categories for flowering phenotypes (**c**), collection site latitude (**d**), and DNA variants in known flowering genes (**e**). Color labels can be found in Supplementary Table 4. **f** Geographic distribution of accessions. Points labeled as: Extremely Delayed Flowering (EDF+): square; Delayed Flowering (DF): circle; Intermediate Delayed Flowering (IDF): triangle; Intermediate Rapid Flowering (IRF): open triangle; Rapid Flowering (RF) and Extremely Rapid Flowering (ERF): star. Colors reflect membership to STRUCTURE groups in **b**. The background map was constructed from Worldclim (http://www.worldclim.org/) elevation date using ArcGIS software (http://www.esri.com/arcgis). **g–j** ML mapping of probable ancestral states for **g** flowering time class, **h** latitude, **i** molecular variant in the *FLT13* gene, and **j** molecular variant in the known flowering regulator *VRN1*. See Supplementary Fig. 5 and 6a for individual line labels and the remaining traits, respectively. **k** Fixed-SNP differences between the three STRUCTURE groups. **l** Median number of non-reference genes added per line from each of the three major groups. **m** Exclusive and shared gene clusters between the three STRUCTURE groups (including admixed lines). **n**, Overlap between core pan-genes in sub-population pan-genomes from the three STRUCTURE groups (without admixed lines) and shell genes in the combined pan-genome. Whiskers in the above plots extend to the most extreme data point which is no more than 1.5 times the IQR

SNP-based ML tree (Figs. 2d and 4a, Supplementary Fig. 7). In addition, PHYML phylogenetic trees of the flowering gene *BdVRN1* created from variants detected by read mapping to the reference genome (assembly v2.0) and variants identified from our genome assemblies were in agreement (Supplementary Fig. 4c,d). Furthermore, the *BdVRN1* gene tree created from the genome assemblies included 75 additional variants that were not detected by read mapping and showed higher resolution and bootstrap support (Supplementary Fig. 4d).

Unlike reference-based strategies, the pan-genome allowed us to examine thousands of PAVs across populations (Fig. 2d). For example, the reference line, Bd21, is the most rapidly flowering line characterized to date, raising the possibility that genes important for vernalization responsiveness may not be present in the reference genome. Indeed, alleles of a predicted CCAAT box-binding NF-YB transcription factor (pan-gene *Brdisv1ABR21022861m*) are present in all EDF+ lines and in some delayed flowering S+ group lines but not in intermediate T+ or rapid flowering lines. In wheat, *Arabidopsis*, and rice, NF-YB paralogs regulate flowering[38–41] suggesting that alleles of *Brdisv1ABR21022861m* may play a role in determining flowering time in *B. distachyon*. NF-Y subunit transcription factors are known to have an elevated rate of duplication, consistent with the presence/absence variation observed in *B. distachyon*[41].

An example of the 1,549 pan-genes that were entirely absent from the reference line is *Brdisv1ABR41022793m*, predicted to encode an ent-copalyl diphosphate synthase (CPS). Neither the reference genome nor any members of the T+ group contain alleles of this pan-gene; however, alleles were observed in all members of the EDF+ clade and in four lines from the S+ group. Pan-gene *Brdisv1ABR41022793m* is slightly more similar to a wheat gibberellin biosynthetic gene (CPS-A[42], 71% identity) than to the most closely related *B. distachyon* reference gene (Bradi2g33686, 69% identity). Flanking genes localize *Brdisv1ABR41022793m* to a block of *B. distachyon* chromosome 2 syntenic to the wheat R-A1 locus[43] on wheat chromosome 3A, whereas CPS-A is reported to reside on wheat chromosome 7A[42]. We compared syntenic genomic regions of six lines that contain alleles of *Brdisv1ABR41022793m* and four lines from which it was absent (Fig. 1d). The presence/absence of the *Brdisv1ABR41022793m* locus was confirmed by analysis of read depth across the region (Fig. 1e). Bradi2g33686 was differentially expressed between dawn and mid-day, similar to the maize gibberellin biosynthetic gene[44], whereas *Brdisv1ABR41022793m* was expressed at similar levels between these time points. These differences in expression suggest different functions for these genes, similar to CPS-encoding genes in other plants[42,45–47].

The sequenced lines in our study vary in resistance to multiple plant pathogens[48–50]. Shell genes may underlie much of the observed variation in resistance since PAVs are common among disease resistance loci in many plants[51]. Non-reference shell pan-gene *Brdisv1Bd1-11011965m* encodes an uncharacterized protein and has alleles in six phylogenetically diverse lines. Its best BLASTP hit is a predicted wheat protein, TRAES_3BF053100160CFD_c1[52] (38% identity). Flanking genes anchored *Brdisv1Bd1-11011965m* to a region of *B. distachyon* chromosome 2, which is syntenic to the wheat *Sr2* locus[51,52]. This locus contains 40 genes, including *TRAES_3BF053100160CFD*, and confers broad-spectrum resistance to wheat stem rust[51] and powdery mildew[53]. The exact gene(s) conferring resistance at this locus have not been determined, but it is linked to extensive PAVs and extreme haplotype divergence among wheat cultivars[51]. In line Bd1-1, expression of *Brdisv1Bd1-11011965m* was induced by wheat stem rust infection. Thus, *Brdisv1Bd1-11011965m* may be involved in pathogen resistance, and the specificity may be controlled by differential gene content as seems to be the case in wheat[51].

**Putative mechanisms leading to gene gain and loss.** There are several, non-exclusive mechanisms for the genesis and elimination of shell genes. As observed in other systems, errors during recombination may create and eliminate genes from the genome[54,55]. Since the loss or movement of non-essential shell genes is expected to be more tolerable than the loss of essential core genes, recombination may cause the distribution of shell genes to be higher in areas of the genome with low recombination. We hypothesize that shell genes may arise via gene duplication followed by sequence divergence as evidenced by their higher non-synonymous/synonymous substitution ratios among lines (Fig. 3d). Transposable elements (TEs) are known to mediate gene duplication and movement[55]; therefore shell genes may be enriched in TE-rich regions of the genome. These mechanisms have been proposed to reduce synteny and homeolog retention after ancient whole-genome duplications within pericentromeric regions, as well as elevate rates of pseudogene formation and transposon accumulation[30].

Although shell genes are observed throughout the genome, the ratio of shell genes to core genes is higher in pericentromeric regions of the chromosomes (Fig. 5a, b). We observed striking relative patterns of polymorphism rate, TE density, synteny, shell/core gene ratio, recombination rate, and TE insertion/excision activity across chromosomes (Supplementary Fig. 3,8,9). The ratio of shell genes to core genes was negatively correlated with recombination rate ($r_s = −0.43$, $p < 3.3e−05$, Spearman's test), and positively correlated with higher density of annotated reference TEs ($r = 0.58$, $p < 4.5e-11$, Spearman's test). Increased shell:core ratio was highly correlated with intra-species TE activity (non-reference insertion and deletion) (Fig. 5b, c; insertion: $r = 0.67$, $p < 4.9e−15$, Spearman's test; deletion: $r = 0.59$, $p < 3.9e−11$, Spearman's test). Linear models predict that shell:core ratio was significantly influenced by intra-species TE activity ($p < 2.2e−16$, McFadden $R^2$, Supplementary Note 3). As our data sets correspond respectively to new insertions in at least one of the 53 lines or to potential excisions from the reference genome, our results are consistent with intra-species TE dynamics mediating a higher rate of shell gene genesis at some locations of the genome. Both TE activity and shell:core ratio were strikingly correlated with the ratio of non-syntenic to syntenic genes in comparisons of the *B. distachyon* reference genome to rice (Fig. 5b, d; $r = 0.84$, $p < 2.2e−16$, Spearman's test, Supplementary Fig. 8). Furthermore, while 74% of core genes were located within DNA segments syntenic to rice, only 24% of shell genes fell into syntenic segments (Fig. 5e). Linear models support the hypothesis that the non-syntenic:syntenic gene ratio is influenced by the shell:core ratio and is a better predictor of reduced co-linearity than TE activity itself ($p < 2.2e−16$, McFadden $R^2$, Supplementary Note 3). Consistent with higher rates of shell gene evolution and location outside of co-linear blocks, shell genes were less likely to have a retained homeolog from past evolutionary whole-genome duplications (Fig. 5f, $\chi^2$-test for independence, $p < 2.2e-16$).

To fully utilize the pan-genome and further elucidate the role of TEs in shell gene evolution, the repeat elements in each de novo assembly were annotated (Supplementary Methods). We quantified gene expression levels across 27 assemblies to evaluate the effect of TEs and other repeat sequences on gene expression. We looked at pairs of allelic genes, defined by synteny of each genome to the reference, with a TE present within 1 kbp upstream of the translation start site in one line but not in another. Genes with a TE within 300 bp of the start site were expressed significantly less than allelic genes without an upstream TE insertion (Fig. 6a, b, Supplementary Fig. 10). On average, TEs were significantly closer to and comprised a larger fraction of the upstream region of shell genes than core genes ($p < 2.2E−16$, Wilcoxon signed-rank test, Fig. 6c, d). Different classes of repeat

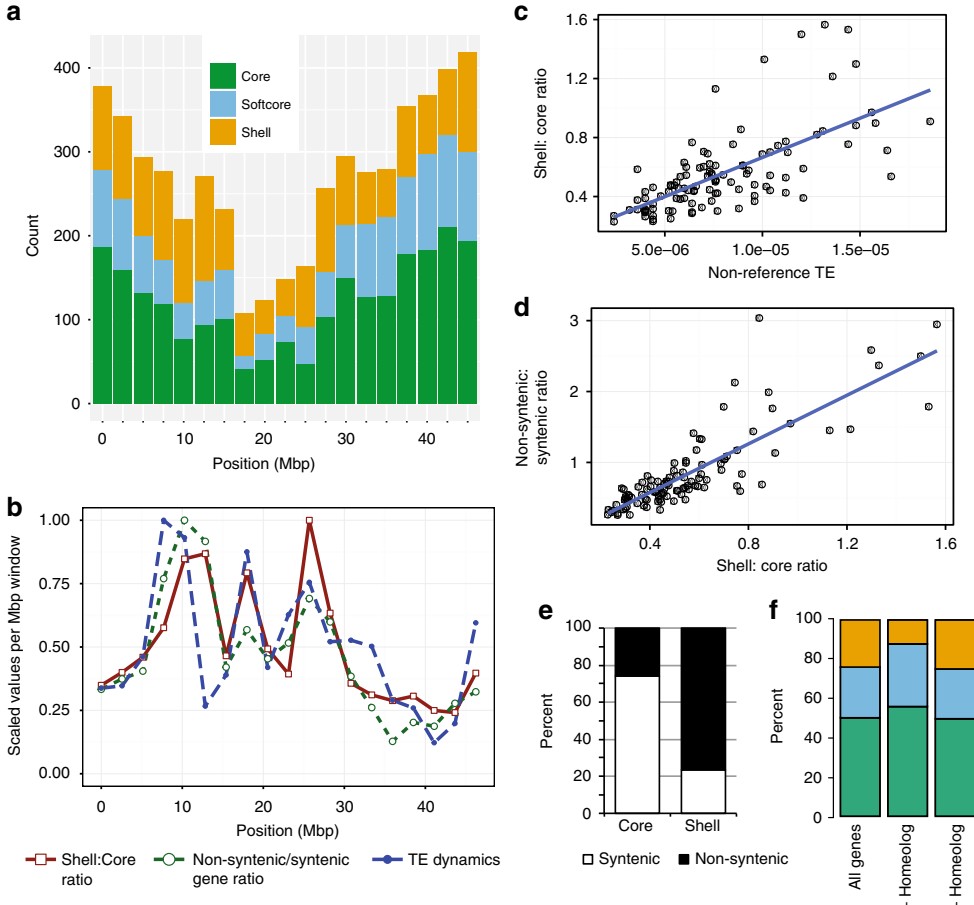

**Fig. 5** Chromosomal characterization of pan-gene subsets. **a** Number of reference genes in respective pan-genome categories within 2.5 Mbp windows along chromosome 4. **b** Shell:Core gene ratio, non-syntenic:syntenic gene ratio (in comparison to rice, *Oryza sativa*), and the number of reference TEs absent from other *B. distachyon* lines compared to the reference (a measure of TE dynamics), plotted for 2.5 Mbp non-overlapping windows along chromosome 4. Frequency of TE "insertion" relative to the reference genome shows a similar pattern as TE "absence" (Supplementary Fig. 9). **c** Intra-species TE insertion frequency vs. shell:core gene ratio within 2.5 Mbp genomic intervals. **d** Plot of non-syntenic:syntenic gene ratio vs. shell:core gene ratio. **e** Percent of core and shell genes in the reference genome that are/are not syntenic with the corresponding rice ortholog. **f** Fewer shell genes in the reference genome have a homeolog that was retained after the ancient grass whole-genome duplication

elements had different effects on the expression of neighboring genes (Fig. 6e). Class 1 elements significantly decreased expression of adjacent genes (binomial test, $p < 0.01$), whereas the decrease associated with class 2 elements was not significant (Supplementary Fig. 10) and Class 1 elements were more likely to be found upstream of shell than core genes (Chi-square test for independence, $p < 2.2E-16$, Fig. 6f). Irrespective of TE class, TE insertions with the largest negative effect on expression were associated with a lower proportion of core genes and higher proportion of shell genes (Supplementary Fig. 10). Furthermore, even when adjacent to a TE, core genes tended to retain their expression level, whereas shell genes tended to be altered in expression (Fig. 6g). These results are consistent with the hypothesis that TE insertion events that alter expression of core genes may be selected against as compared to shell genes. As TEs are known to mediate gene transposition and removal, the higher frequency of retained TE insertions adjacent to shell genes may mediate their dynamic behavior among individuals of the species.

## Discussion

The primary goal of our study was to accurately estimate the size of a plant pan-genome. As mentioned in the introduction, the challenges of producing numerous, complete de novo genome assemblies have prompted previous plant pan-genome studies to use approaches that avoid whole-genome de novo assemblies (e.g., reference-based with targeted assemblies, pan-transcriptomics, metagenomics) or examine small numbers of lower quality assemblies. These studies likely underestimate the size of the pan-genome because they have limited power to detect novel contiguous sequence outside the reference genome or they use a small number of highly fragmented assemblies of unknown completeness. Nevertheless, they provide important insights into functional aspects of plant pan-genomes. For example, reference-based approaches found that non-reference genes are often involved in processes related to traits of agronomic interest such as environmental stress and plant defense responses and may be implicated in heterosis[9,10,15,16,56]. They have also shown that as much as 30 percent of genes in a reference genome may be affected by PAV[10,11,57]. Studies focused on de novo assembly approaches have observed similar functional attributes of genes associated with PAV across the species-wide pan-genome[5,14]. These studies have also shown that up to 16% of the species-wide pan-genome lies outside the reference genome[13]. In pairwise comparisons, maize inbred lines show extreme PAV and copy number differences, which if extrapolated would lead to an immense species-wide pan-genome for this species[58,59] suggesting that maize pan-genome size estimates represent a lower bound on pan-genome size.

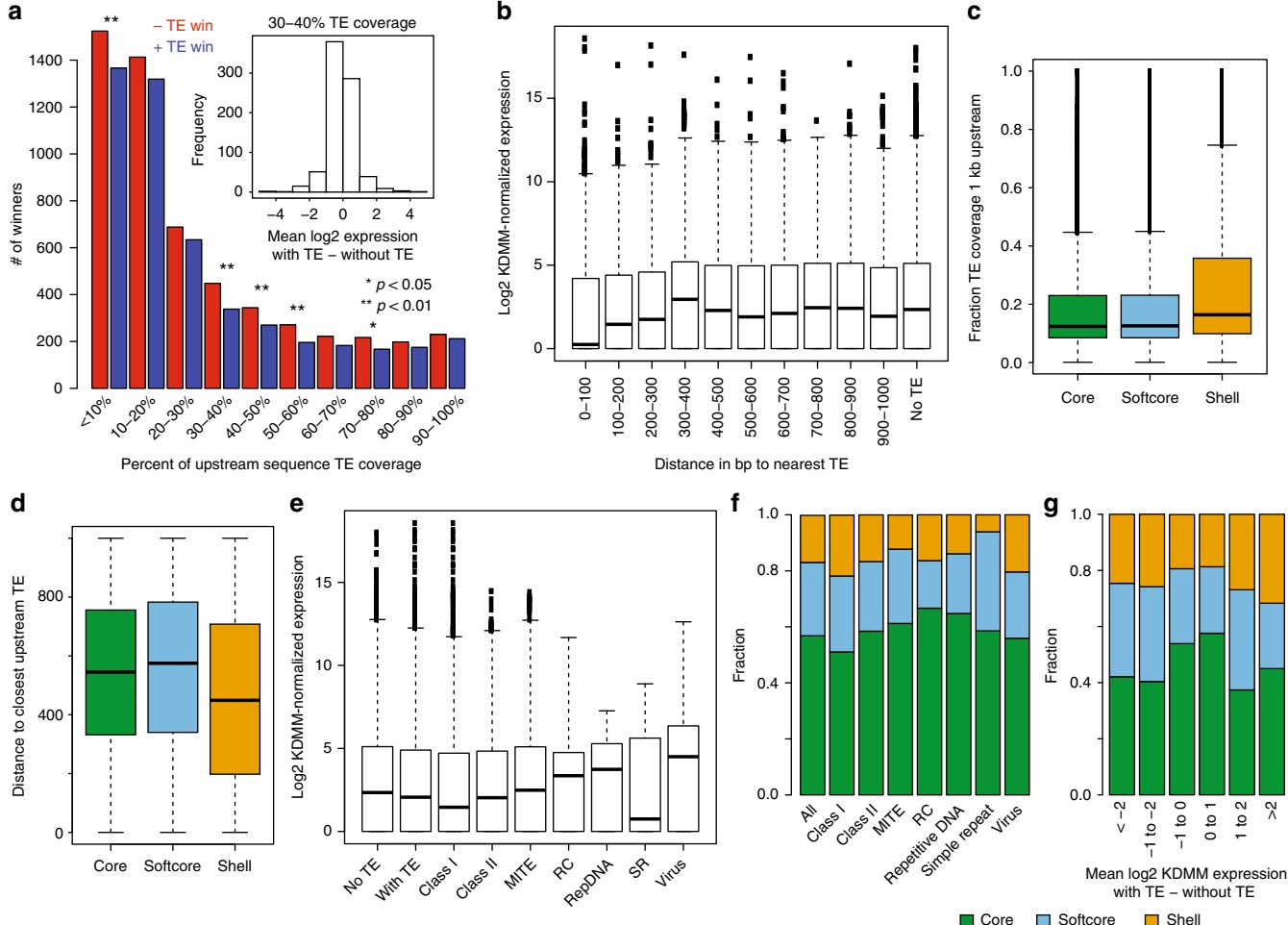

**Fig. 6** Effect of TE insertions on gene expression. **a** Number of winners in binomial distribution sign tests (horserace comparisons) of expression level among alleles with/without TEs, p-values based on a binomial test. Inset shows log2 expression differences between alleles with/without TEs, for focused comparisons of alleles having 30–40% TE coverage. **b** Expression level of genes with TE within specified distance to the translation start site. **c**, Fraction of TE coverage of 1 kbp upstream of core, softcore, and shell subsets. **d** Distance to closest upstream TE for core, softcore, and shell subsets. **e** Expression level of genes adjacent to respective TE classes. **f** Percent of genes adjacent to repeats of respective classes assigned as core, softcore, and shell categories. **g** Genes binned according to mean log2 expression difference between alleles of a gene with/without TEs, each colored according to membership in core, softcore, and shell categories. Whiskers in the above plots extend to the most extreme data point which is no more than 1.5 times the IQR

By including assembly controls and conducting extensive cross validation of PAVs via read mapping and phylogenetic analysis, our study provides strong evidence that the *B. distachyon* pan-genome is considerably larger than the genome of any individual plant of this species. Indeed, the high-confidence pan-genome contains 7135 pan-genes that were not contained in the reference genome and nearly half of the genes in the high-confidence pan-genome are found in only a subset of lines. The completeness and number of de novo genome assemblies utilized in our study, the selection of lines to maximize the sampled genetic diversity, and the fact that *B. distachyon* is a wild plant that has not experienced a domestication bottleneck all contributed to the large pan-genome observed for this species. Our estimated size of the *B. distachyon* pan-genome is a conservative estimate due to collapsed tandem repeats, co-clustering of related paralogs within an assembly to the same pan-gene, and lack of gene annotations for novel genes lacking expression or homology support (required for our gene annotation). Thus, we have defined a lower bound for the size of the *B. distachyon* pan-genome. Precisely defining the core genome becomes more problematic as the number of genomes sampled increases due to the challenge of assembling and annotating a particular gene model correctly in every sequenced

genome. Our softcore category reflects this uncertainty. It should be noted however, that uncertainty about the exact set of core genes does not affect our estimate of pan-genome size, our primary objective, because the pan-genome is simply the sum of the genes in the reference genome and all the non-reference pan-genes. While the *B. distachyon* pan-genome is larger than the pan-genomes reported for several other species, it is not completely unexpected based on results from other species. For example, an estimation of the maize pan-genome based on the pan-transcriptome[16] was of similar magnitude to the *B. distachyon* pan-genome. In addition, a study of 10 reference-guided assemblies of *Brassica oleracea* found that 20% of pan-genes showed PAV[10] which is consistent with our simulation of the pan-genome size for 10 *B. distachyon* lines (Supplementary Fig. 2b). Similarly, true de novo assemblies of 15 Medicago genomes found that 42% of genomic sequence was found in only some accessions[13].

Previous studies utilizing a purely de novo assembly strategy have focused on relatively small numbers of less complete assemblies making it difficult to ascertain the effect of population structure on pan-genome size and phylogenetic distribution. Powered by the much larger sample size in our study, we show

the importance of population structure in elaboration of the species pan-genome. In our study, PAV correlates with phylogenetic relatedness and pan-genes that are non-core in the species-wide pan-genome are often core within sub-populations. After excluding admixed lines, the three major population groups in *B. distachyon* differ greatly in their complement of pan-genes. Hundreds of pan-genes are core to one sub-population while not found in other populations (Fig. 4n). Sub-population core genes may in fact contribute to perpetuating population structure. For example, we identify a putative NF-YB transcription factor in all EDF+ lines that may potentially contribute to temporal differences in onset of flowering relative to non-EDF+ lines that could reinforce the genetic distinctness of the EDF+ sub-population via pre-mating reproductive isolation. Genes that appear dispensable within the species-wide pan-genome may in fact be extremely important for the biology of a sub-population[60]. Individuals from previously non-sampled sub-populations contribute far more to increases in pan-genome size than the addition of closely related individuals. This underscores the importance of careful selection of individuals for pan-genome studies, particularly in species like *B. distachyon*, in which geography may be a secondary factor in shaping population structure, and autogamy may influence the spread of and selection on PAVs. The degree of population structure within a species and genetic bottlenecks, such as domestication, need to be taken into account when interpreting pan-genomes and may be reasons some crop plants may have smaller pan-genomes than wild species such as *B. distachyon*. Despite the large overall size of the *B. distachyon* pan-genome, individuals in our study share the vast majority (90% on average) of their genes at the pairwise-level. Previous studies have observed greater amounts of pairwise PAV between individuals in other plant species[58,59,61]. Thus, the amount of PAV that we observe in the compact genome of *B. distachyon* is likely to be dwarfed by PAV in larger, more complex genomes such as maize[58].

The functional enrichments observed in the shell genes, their expression levels and patterns, and their high evolutionary rates are consistent with a scenario in which shell genes evolve rapidly and are more likely than core genes to be adaptive under certain environmental conditions. For example, we found that shell pan-genes are enriched in RIPs, which can provide an advantage in the presence of pathogenic fungi or herbivorous insects. In contrast to higher frequency pan-genes in the shell compartment, the long tail in pan-gene frequency in Fig. 2a may indicate that a large portion of PAVs is at low frequency and may be under negative purifying selection. This fits a scenario where new pan-genes are continually created and lost unless they are adaptive under some conditions. Selection against indels has been noted in other systems[61]. The higher relative abundance of shell genes in non-syntenic blocks of the genome and the higher level of intra-species TE insertions and deletions near shell genes suggest TE dynamics as an important mechanism for shell gene creation and removal, similar to the role of TEs in generating inter-species differences in gene content[30,52]. In light of our study, this concept may be expanded to intra-species variation as previously speculated[55]. Our observations are consistent with previous results in *Glycine soja*, where a higher level of PAVs in pericentromeric regions was noted[14]. Class I elements were proportionally overrepresented upstream of shell genes. This would be compatible with retrotransposon-mediated long terminal repeat (LTR)–LTR illegitimate recombination as a mechanism by which shell genes are lost/gained. Indeed, it has been suggested that LTR recombination actively counters retroelement expansion in *B. distachyon* and may partly explain its relatively low complement of repeats and small genome size[22]. In contrast, it has been proposed that retroelements persist for very long periods of time in the closely related Triticeae, which have high repeat content

and large genomes. High repeat content complicates genome assembly and thus precludes non-reference-based pan-genome analysis. Nonetheless, advances in technology may soon enable these analyses in large genomes with higher repeat content such as maize, which is believed to have a large pan-genome of unknown size[8].

The individual genome assemblies, associated data sets as well as interactive tools for mining the *B. distachyon* pan-genome are available at the BrachyPan website (https://brachypan.jgi.doe.gov/). These tools, in combination with the experimental resources available for *B. distachyon*, allow further investigation of the mechanisms and functional consequences of intra-species gene dynamics.

## Methods

**Plant germplasm, DNA extraction and sequencing**. The sources of the lines used in this study are described in supplementary table 1. High molecular weight nuclear genomic DNA was isolated from 10–20 g of leaf tissue collected from 4-week-old seedlings using a nuclei isolation protocol[62]. DNA was randomly sheared into ~250 bp fragments, and then used to create Illumina libraries. Sequencing was performed on Illumina HiSeq2000 and HiSeq2500 sequencers, generating 73–100 bp paired-end reads for the 54 *B. distachyon* inbred lines at 92x median genome coverage. Also, 4 kb mate-pair large-insert libraries were constructed for eight lines. Illumina short reads were processed by the Joint Genome Institute read filter (rqcfilter, https://sourceforge.net/projects/bbmap/) to remove common contaminants, adapter sequences and low-quality reads.

**De novo genome assembly and annotation of 54 inbred lines**. The eight genomes for which we had appropriate fragment and mate-pair libraries were assembled with ALLPATHS-LG[63] and the remaining genomes were assembled with Velvet[64] (v1.1). As a control, we assembled Illumina sequence data from the same line used to create the reference genome, Bd21. To make the assemblies easier to work with and provide physical context, we used synteny to the reference genome to order and orientate scaffolds into five pseudomolecules corresponding to the five *B. distachyon* chromosomes and additional super-scaffolds containing unassigned sequence. Significantly, no reference sequence was added to any of the assemblies during this process. Details on genome assembly, gene and transposable element annotation and analysis can be found in Supplementary Methods.

**Clustering pan-genes**. As annotated genes residing on each individual assembly were not necessarily syntenic, we identified related genes across assemblies by grouping the 1,796,495 gene models across assemblies by sequence similarity using Markov clustering in the GET_HOMOLOGUES-EST pipeline[17] (https://github.com/eead-csic-compbio/get_homologues) with minimum alignment coverage of 75% (Supplementary Methods). The resulting clusters were divided into cloud, shell, soft-core and core subsets based on the number of lines contained in each cluster.

**Variant calling and high-confidence SNPs**. Variants were identified from BWA alignment of short reads to the *B. distachyon* reference genome followed by analysis with SAMtools mpileup to detect variants that passed initial quality filtering by vcftools (v0.1.12b) vcf-annotate (defaults+3× average depth cutoff) resulting in 5,994,487 variants. These variants were used in the gene tree for *VRN1* and its comparison to the equivalent tree based on the assembled genomes.

**Phylogenetic analysis and flowering time measurements**. For whole-genome phylogenetic trees the initial set of variants was further filtered to retain only SNPs that were supported by at least three reads in every individual, and had an unambiguous genotype for at least 52 of the 53 lines. After filtering we were left with 3,933,264 high-confidence SNPs that were used for phylogenetic analysis. For whole-genome phylogenetic analysis, maximum Likelihood (ML) phylogenetic analysis was performed on the high-confidence SNPs of the 53 *B. distachyon* lines in RAxML[65] (v. 8.0.0). The 3,933,264 SNP sites were proportionally distributed across the five chromosomes of *B. distachyon* (see Supplementary Methods). Growth conditions and scoring of flowering time are described in the supplementary Methods.

**Analysis of flowering time and related molecular traits**. The evolution of the 25 flowering time traits and related molecular traits recorded in 53 sequenced *B. distachyon* lines was analyzed using BAYESTRAITS v 2.0[66] (Supplementary Methods).

**Transposable element insertions and deletions relative to the reference genome**. Transposable element (TE) variants relative to the reference genome were inferred with TEMP[67] (v1.05), using TE consensus sequences from the TREP

database (http://botserv2.uzh.ch/kelldata/trep-db/index.html). Briefly, for each accession, TEMP uses breakpoints of homology between paired-end sequences to detect TE insertions present in the sequenced accession but absent from the reference genome (non-reference insertions). In addition, TEMP uses deviations from the expected insert size to identify TE insertions present in the reference genome but absent from the sequenced accessions (absent insertions). Only suspected TE insertions/absences were retained which were supported by at least two read pairs from each side. The filtered output was then used to create TE insertion and absence matrices, respectively.

**Gene expression studies**. Expression profiles for leaves from 36 accessions was conducted using 3′ tag sequencing as described in the Supplementary Methods. Expression analysis of accession Bd1-1 during its interaction with the grass fungal pathogen *P. graminis* f. sp. *tritici* (*Pg-tr*) was conducted using 101 bp single-end Illumina sequencing as described in the Supplementary Methods.

**BLASTP similarity of non-reference genes to other proteomes**. Peptide sequences corresponding to non-cloud non-reference genes were queried against the NCBI nr database (release 11 July 2014) with BLASTP to identify best matches to other species with *E*-value ≤ 0.0001, if there was one. The mean query coverage of retrieved hits was 77.8%.

**Synteny between the reference genome and related grasses**. We used an in-house synteny pipeline, comprised of all vs. all BLASTp between respective proteomes as well as self-BLASTp, using a HSP filter of 1E−5, and assignment of putative orthologs using Cscore and four-fold degenerate transition/transversion rates to identify syntenic genes. Adjacent (co-linear) gene matches were merged into syntenic segments with a maximum allowed separation of five non-co-linear genes. We required four co-linear gene pairs within merged segments in order to retain respective segments.

Additional experimental details can be found in the Supplementary Methods.

**Data availability**. The sequence assemblies, gene annotations and related information can be downloaded from the project website: https://brachypan.jgi.doe.gov/. The raw reads for the genomic sequences are available (Supplementary Table 1 for GOLD biosample identifiers and SRA information). Seeds for the lines used in this study are available from the USDA NGPS or by request from the authors.

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

## Acknowledgements

The work conducted by the US DOE Joint Genome Institute is supported by the Office of Science of the US Department of Energy under Contract no. DE-AC02-05CH11231. D.P. W. and R.A. were funded in part by the National Science Foundation (grant no. IOS–1258126), and the Great Lakes Bioenergy Research Center (Department of Energy Biological and Environmental Research Office of Science grant no. DE–FCO2–07ER64494). TEJ and DLDM were supported by NSF PGRP grant IOS-0922457. We thank Jason Stajich for advice on dN/dS software. P.C. and B.C.M. were funded by Spanish MINECO (CGL2012-39953-C02-01 and CGL2016-79790-P). B.C.M. was partially funded by DGA—Obra Social La Caixa (grant number GA-LC-059-2011) and Spanish MINECO (AGL2013-48756-R, CSIC13-4E-2490). PC was partially funded by Spanish Aragon Government-European Social Fund (Bioflora). BCM and PC thankfully acknowledge the resources from the supercomputer "Memento" and assistance provided by BIFI-ZCAM. M.F. acknowledges the University of Minnesota Experimental Station USDA-NIFA Hatch/Figueroa project MIN-22-058.

## Author contributions

S.P.G. designed experiments, collected and analyzed data, integrated data sets, wrote the initial draft; B.C.M. conducted gene clustering, analyzed data and conducted pan-genome size simulations; D.P.W. and R.A. conducted flowering time experiments and provided input with evolutionary analyses; D.L.D.M. and T.E.J. provided and analyzed expression data; D.B. and M.F. analyzed effects of TEs on expression; S.S. annotated the genomes and looked for missed pan-genes in the reference; C.S. and A.R. analyzed the loss/gain of TEs; W.S., J.M., and A.L. detected variants with respect to the reference genome; L.T. helped initiate the project, select lines and prepare DNA; N.D. and K.G. did the VRN1 analysis; JP helped create gene families; K.B. managed the genome sequencing project; H. B. provided germplasm and helped initiate the project; A.L.C. provided germplasm and conducted some population analysis; D.G. and P.D. designed and created the BrachyPan website; L.M. provided germplasm and DNA; M.F. conducted pathogen experiments and provided expression data; P.C. conducted population and evolutionary analyses of the genomic and flowering data; J.P.V. conceived the project and overall approach, coordinated projects, interpreted data and revised the manuscript. All authors contributed to and revised the manuscript.

## Additional information

**Competing interests:** The authors declare no competing financial interests.

