## [Peer Review File · Nature Communications]

Reviewers' comments:

Reviewer #1 (Remarks to the Author):

The authors perform a pangenome study using 54 accessions of a grass species *Brachypodium distachyon*. They perform pangenome assembly, annotation and functional analysis, attempting to link some of the variable genes to phenotypic traits. *B. distachyon* is an undomesticated plant used as a model plant species and as such is of interest to wide plant research community.

Overall, the methods used are appropriate and mirror design of similar studies in other species. The authors compiled a large dataset. The manuscript describes a significant advance in our understanding of this species as well as plant genomes in general but could be improved by additional validation and removal of some of the more speculative sections.

The majority of conclusions can already be found in manuscripts describing pangenomes of other species and they simply confirm similar observations in *B. distachyon*. The manuscript could be improved by detailed comparison with these previous studies, additional functional analysis, especially in relation to the population structure and geographical location of accessions.

The title is rather vague and perhaps misleading, I would recommend naming the species. The link to phenotypic variation is rather speculative and requires further support.

I agree that the small genome size makes it a relatively easy model for pangenome analysis, however the authors should consider how well findings translate to species with larger genomes, specifically those with greater TE content. One of the weak points of this manuscript is the lack of comparison with other pangenome studies. While the analysis of the *Brachypodium* data is mostly strong, without putting these findings into a broader context limits their interpretation.

The main area which could be improved is in the validation of the assemblies and their annotation. Variation in assembly quality and annotation can lead to incorrect predictions of gene presence/absence variation and additional evidence supporting the gene variability presented here is required before conclusions can be confidently made. The authors acknowledge that 5 assemblies are of low quality and were removed from the analysis, but the quality of the other new assemblies is variable and is likely to impact on interpretation of gene loss.

On page 3, it is suggested that some genes and control elements are truncated. It would be valuable to quantify these and discuss in relation to genome assembly quality.

A very small number of genes were found in all lines. This contrasts with what is observed in other pangenome studies and suggests that some assemblies may be of poor quality. More details of gene loss across the lines are required – for example, are some lines missing many more genes than others. Was there any relationship between ‘lost’ genes and repeat motifs?

The terminology of core, soft-core, cloud etc. is new and seems arbitrary. Without some biological basis for this classification, I suggest using the more standard core and variable gene terminology.

The cloud genes were removed as likely artefacts, but some were then identified as real and expressed. Presumably it is possible to model the pangenome expansion with increasing number of lines to predict the true number of genes which are relatively rare in a representative population.

The shell pan genes which are missing in some lines but not others require further validation such as PCR to ensure they are not artefacts.

Genes with repetitive motifs tend to collapse in short read assemblies. It would be valuable to assess whether the variable genes had a greater frequency of repeat motifs. This is not clear from the current high level annotation presented.

The link between variable genes and disease resistance is highly speculative and needs much stronger support to be included. Disease resistance genes contain repeat motifs and while many do show presence/absence variation, their loss is also likely to be an artefact of assembly or annotation.

The location of variable genes on pseudomolecules required detailed comparison with observations in other species.

The wording of the association of variable genes with TEs is misleading, while I do not disagree with the observation that they are correlated, there is no evidence for one ‘influencing’ the other. TE movement may well be associated with gene movement and loss, but this is not demonstrated in this manuscript. The analysis of TE elements in relation to PAV is interesting, however additional details would make it much stronger. Could we find out more about types of TE elements found in vicinity of variable genes? Are some of the more likely to mediate PAV than others?

It is understood that wild germplasm maintain traits which may be of agronomic value so this suggestion is not new and several examples are published. The significant number of variable genes associated with telomere annotation requires an explanation, could this be an artefact due to repetitive motifs in these genes?

Two assemblers were used, was there any difference between gene content that could be associated with these different assembly processes? I am surprised that the 75bp reads from the HiSeq2000 produced such poor assemblies compared to the 100 bp reads. Is this more to do with quality of the older reads than the length?

CEGMA is now replaced by BUSCO, though it would be valuable to include both analysis.

It looks like RNASeq was generated for 36 accessions, but not used in the annotation. Using RNASeq could improve annotation of accession-specific genes not found in Bd21, Bd21-3, Bd1-1.

The authors perform population analysis of the accessions, however much more could be done for this analysis. The analysis splits accessions into three clades based on SNPs. A similar clade structure was recovered based on PAV analysis. Authors do perform analysis of flowering time traits. However any further analysis of PAVs in relation to population structure is lacking. A more in depth analysis would allow novel insights into evolution of undomesticated plant species. What are the functions clade specific genes? Can those be linked to geographical locations? Do the relationships recovered based on PAVs and SNPs mirror each other completely?

I am not sure what do the authors mean by this: The first clade contains lines from multiple geographic locations that almost all exhibit an extremely delayed flowering (EDF+) phenotype (Fig. 2c, 4b, Supplementary Fig. 4), indicating that flowering time is a major factor in the divergence of populations (Fig. 2c, Supplementary Table 3). Underscoring the relationship between flowering time and the pan-genome. Bayes factor (BF) tests for potential correlated evolution of flowering time traits (Fig. 4c) and molecular variants in genes of known flowering regulators (Fig. 4e) revealed strong associations (Supplementary Fig. 5, Supplementary Tables 3, Supplementary Information) What exactly is the proposed relationship. Could the authors elaborate and perhaps put into the context of what is currently known about the role and function of specific flowering time genes?

Reviewer #2 (Remarks to the Author):

The authors find a surprising amount presence-absence variation (PAV) for genes in the plant genus *Brachypodium* and characterize how such variable genes differ as a class. The authors further argue that at least some these PA polymorphisms may contribute to functionally important phenotypic variation.

Overall, I think the work is sound and is certainly substantial (there is, in fact, too much data presented to even discuss it all in the main MS or for me to give a thorough review of it all).

It is significant in demonstrating that a single reference genome can give only an incomplete enumeration of the gene catalogue for a eukaryote, expanding the application of the “pangenome” from its more customary domain of microbes. Further, the findings raise the possibility that the variation in gene content among lines may underlie functionally important phenotypic variation (but see the caveat below).

Major comments

1. (Needs to be addressed by revision) The authors assert that the three groups of lines should be considered a single species, and thus share a single pangenome, based on the fact that crossing in the laboratory is possible and admixed lines exist in nature (line 234). However, the latter are rare, and crossability between distinct grass species is common, including even intergeneric crosses. The F_{ST} values in pairwise comparisons range from 0.42-0.63, which suggests considerable genetic isolation. The EDF+ group, in particular, flowers late, a strong pre-zygotic reproductive isolation barrier, and shows 100-150,000 fixed SNP differences, though admittedly the case for S+ and T+ being distinct is weaker (Suppl Fig 4). The exact taxonomic rank of these groups is not important, but it does suggest analyzing the PAV variation in a hierarchical manner, since patterns of presence and absence in a gene may not be the same in all subtaxa.

This raises some interesting questions:

- a. Is a gene classified as shell gene among all subtaxa core to one or two of them?
- b. What is the differentiation among the taxa in gene content?
- c. Is the ratio of pangenome to reference smaller when the pangenome is constructed for each subtaxon separately?

2. Related to that, It would be helpful to have a clearer visualization of the matrix of presence-absence variation of all the genes across all the lines than the current Fig 2c. A hierarchical clustering of the genes (and possibly the lines) in Fig 2c based on similarity in presence-absence vectors might make the patterns more evident, and illuminate, for instance, if the shell genes are distributed evenly among the subtaxa.

3. The findings regarding phenotypic effects for Brdisv1ABR41022793m and Brdisv1Bd1-11011965m are very interesting and highly suggestive, but are not conclusive in linking phenotypic variation to PAV. The assertion in the title strikes me as too strong. If the authors still

wish to make this one of the primary take home messages of the paper, I would at least suggest moving Suppl Fig 5 into the main MS, beefing up the superficial description of these results in the main MS, and making the case for the overall conclusion, including caveats, more explicit in the Discussion.

4. The authors conservatively exclude “cloud” genes from the analysis, and distinguish the “soft core” from both the “shell” and “core” classes, both of which seem sensible. But the lack of a soft-shell seems a bit arbitrary given the distribution shown in Fig 2a – there is a sizable shoulder of genes present in 2, 3, 4, 5 lines, and hardly any found at intermediate frequencies. If the authors prefer not to include a soft-shell category in a reanalysis, I would at least like to see a justification for the arbitrary cutoffs in the Methods, and an explanation in the Discussion about how choosing different cutoffs/categories would affect the interpretation.

5. It would be helpful to the reader to have more in the (unusually brief) Discussion comparing the general pattern of PAV seen here to what’s been seen, or may be seen, in other systems. For instance, our group has demonstrated that there is purifying selection against gene deletions that are variable within a natural plant population (e.g. doi:10.1093/gbe/evt199, not cited).

6. Why would a maximum likelihood tree that forces bifurcations and does not allow reticulations be appropriate for modeling the evolutionary history of these lines? This is not a critical problem for this MS, since the interpretation of tree is incidental to the main conclusions. But in general, where there is ongoing mating and recombination, a phylogenetic tree is not the correct model for explaining similarities among genotypes, and is potentially misleading in its interpretation. A simple hierarchical clustering method would be more appropriate (see also point 5).

Minor

7. Include the estimates of F_{st} within the main MS rather than the Supplementary Methods.

8. The Supplementary Methods contain quite a number of results, so the name is a little misleading.

9. I am not familiar with the term “base perfect” and don’t see an explanation in the cited reference (Ref12).

10. Spell out “FPKM” upon 1st occurrence.

11. Is the legend in Fig 5b all correct? The text implies that TE abundance, shell:core ratio and nonsyntenic:syntenic ratio are positively correlated. But the latter two appear to be correlated with “TE absence” here.

12. I’m not sure I see the trend in Fig 6f that the authors are reporting.

13. Line 779: wouldn’t the minimum BLAST coverage be more informative than the mean?

14. Line 812: define or provide a reference for “Cscore”.

Typos and such

15. Fig 6f: legend color for 'shell' is pink not orange
16. Line 628. Is Ref15 the correct citation?
17. Line 640L "miss-annotated" -> "mis-annotated"
18. Line 652. "R2" -> "R^2"
19. Line 657 Provide figure numbers.
20. Line 983. Capitalize "Gene Ontology".

Signed, Todd Vision

Reviewer #3 (Remarks to the Author):

The manuscript from Gordon et al "A plant pan-genome links extensive variation in gene content to phenotypic variation" deals with a large pangenomic approach in the wild grass *Brachypodium distachyon*.

The study is very interesting of of high importance in those years of population genomics and GWAS, and provides a lot of clues that can explain unexpectd results from reference-based approaches.

The experiment it self is well conducted, with a lot of supporting data and analyses. The paper is well written, with for me few corrections, mainly deeper informations needed rather than missing ones (see below).

My two main questions is on the origin of the genes from what the authors called the shell and cloud genomes and on their potential impact on speciation.

In page 4 and 5, the tried to identify a potential origin through duplication or drift from reference genes, and succeed for some, but not all. I would like to know if authors, even for discussion purpose, have any idea of the origin of those non-duplicated gene, ie of the neogenesis in this specific case: horizontal transfer ? Neogenesis per se (as in papers from Long, such as Evolution of New Genes, 2015. Oxford Bibliographies or Kemkemmer C, Long M (2014). New genes important for development. EMBO Rep doi: 10.1002/embr.201438787) ?

This is a tremendous subject that may explain 'rapid' adaptation of organisms to a wide range of environments.

In the same way, speciation is known to be at least partially linked on recombination issue in zygote formation or in meiosis errors in hybrids. Thus, is there any clue about the impact of such pan-gene difference on the recombination/meiosis, such as difficulties to obtain a 100% fertile descent in crosses between individual having highly different dispensable/shell genome ? It could be interesting to discuss this issue, but perhaps it is a too big one for the current scope of the paper.

As other requests/corrections/point of discussions:

- 1- Why authors use the terms shell and cloud genomes instead of the generally used dispensable and individual ones ?
- 2- p3, l 118-119: what is the median length of added segments ?
- 3- p4, l140: how did you limit the soft-core/shell frontier ?
- 4- p4, l144-145: not sure that most cloud gene are artifact... Did you check them for Pfam or other system thoroughly for verifying their potentiality ?
- 5- p4 l148-149: 22% of them are expressed in leaves (so many more in the whole life of the plant). What is the relative part of core genes expressed in leaves ? This could be an indication of the true number of real cloud genes (individual specific ?)
- 6- p4 l160-166 and latter on: all your synteny outside the Brachypodium are based on one, max 2, genome per species, and thus may be also biased in terms of conclusions. i would have been less strong in conclusions about their relatives in other species. Perhaps they exist but we did not find them yet. This could be tested e.g. on rice using the RPAN data (<http://cgm.sjtu.edu.cn/3kricedb/>), but it is not mandatory here.
- 7- p6 l238: where are the copy number (CNV ?) analyses ? I found PAV, SNP but not CNV there. Mistypo ?
- 8- p7 and generally: are they some genes that are never associated (ie antagonist?)?
- 9- p8 For TE analysis, is there any link with the TE size and nature/level of expression of genes ? I mean if TE are linked to shell genes, normally they would be more recent copies, thus longer than background ones ?
- 10- p8 core genes, if they are essential, will be expected to be more transcribed than shell one...

Supp Methods:

- 11- in general, an effort must be done there to homogenize the writing. Please correct the version of software (sometimes given, sometimes not), the abbreviations (given sometimes after having used them), the text in itself (p17, l520-521 text is already written above).
- 12- p17 l530-531: 5 to 25% of sequences cannot be mapped (234kp relative to 1Mb genes): I suspect it is mitochondrial/chloroplastic data ?
- 13- p19 l607: why having worked on 27 lines only ? Are they representative of diversity ?
- 14- p19 l648: please provide the correct link.
- 15- p20 l668: it finished quite abruptly...
- 16- p20 l684-686: the selection is based on alphabetical order and thus will be biased in direction of a certain type of diversity ?
- 17- p22 Pangenome size simulation: can you estimate the minimal number of genomes to be sequenced to close it with your data ? It is not really clear...
- 18- Supp figures 6k-0 are completely unreadable...

In conclusion, I think the paper is of high interest and must be published as soon as those questions and corrections will be addressed

Francois Sabot

Reviewer #4 (Remarks to the Author):

Gordon et al., A plant pan-genome links extensive variation in gene content to phenotypic variation

The authors embark towards sequencing and comparative analysis of 54 different cultivars of *Brachypodium distachyon*, an important model organism for grasses (and in particular cereal grasses) in general. The manuscript is well structured and very appealing to read and to a large extent I've been impressed by the clear and straightforward analysis, the very good documentation in the supplementary material and the nice and illustrative figures.

While the pan-genome has been an issue that has been well addressed in bacterial genomes thus far (and with few exceptions) this hasn't been a target for plant genomes in general.

Economics of genome sequencing and resequencing now allow to address this question also in larger eucaryotic genomes. The authors used a very nice and illustrative classification schema and dissected the genome(s) into core, shell and cloud constituents with the conclusion that lots of additional sequences and genes are found (or not found) in different cultivars. Structure analysis subsequently subdivides the different cultivars into early flowering and cultivars that stem from Spain and Turkey. Some criticism here: Looking at figure 4f and the localities of where the cultivars have been collected and the STRUCTURE analysis this becomes somewhat trivial. Collection areas are geographically separated anyhow....

However the analysis is to a large extent very robust and well documented and I can envision that future pangenomic analysis in various plant genomes might use the *Brachypodium* approach as a blueprint, a motivation and an inspirational resource

Criticism:

only few but...

line 116: "...and did not contain any 21 bp sequence found...." This is really enigmatic and even from crosschecking the supplementary material it doesn't become very clear. Well since this reviewer has some background in bioinformatics and genome assemblies I can imagine what this means but for the less expert readers it might help to render this a bit more understandable

- I am not clear on whether any efforts were undertaken to filter for potential contaminants. It would be easy to increase a pan-genome by all kind of bug sequences....

- what kind of genes are in the shell and cloud category? Did I miss this?

- numbers: the authors are talking about 61155 pan genome clusters. A striking number that doesn't seem to fit to the numbers you give in figure 4i and would translate in a massive inflation of genes given that the reference genome has "only" 31000 genes or so. Please clarify
- line 178/179: 80% sequence similarity is used as a cut-off to distinguish or speculate about functional divergence. Where does this value come from? It certainly depends on how deep you go in the functional conservation and regulation interferes as well... well a broad field.
- line 199 ff and figure 3d and 3e: is there a measure of statistical significance
- line 201: "shell genes are also are less likely..." remove one "are"
- line 203: "...core genes are expressed at higher levels and are more broadly expressed..." How is broadly defined? I understand as a biologist but I wonder about the definition used in mathematical terms.
- line 236: To be honest I don't get the admixture argument for Arn1 and Mon3. Either my printout is of bad quality or it looks (in tendency) very similar to the neighbouring cultivars (fig 4b)
- line 363 ff: "...some shell genes in crop plants may encode traits of agronomic value..." Well maybe yes, maybe no.... After all Brachypodium is not a crop plant and even if this is "only" a speculation in the discussion I don't see a value in this. Leave out (?)
- the BrachyPan website and availability of data: when I checked the website it was "temporarily unavailable". Please fix and make sure the data become publically available (along with a potential publication). I have to say that so far my experiences with JGI and JGI data release policy was always very positive but I've seen very bad examples as well. Certainly not wishworthy

Response to reviewers

Reviewers' comments:

Reviewer #1 (Remarks to the Author):

The authors perform a pangenome study using 54 accessions of a grass species *Brachypodium distachyon*. They perform pangenome assembly, annotation and functional analysis, attempting to link some of the variable genes to phenotypic traits. *B. distachyon* is an undomesticated plant used as a model plant species and as such is of interest to wide plant research community.

Overall, the methods used are appropriate and mirror design of similar studies in other species. The authors compiled a large dataset. The manuscript describes a significant advance in our understanding of this species as well as plant genomes in general but could be improved by additional validation and removal of some of the more speculative sections.

The majority of conclusions can already be found in manuscripts describing pangenomes of other species and they simply confirm similar observations in *B. distachyon*. The manuscript could be improved by detailed comparison with these previous studies, additional functional analysis, especially in relation to the population structure and geographical location of accessions.

We disagree with the assertion that our work simply confirms observations in other species. To our knowledge, there is no published pan-genome that includes more than seven individuals, and the quality of those assemblies was an order of magnitude lower than that presented here. All other published works with larger numbers of individuals use reference-based, meta-genome, or transcriptome based approaches. While most of the plant pan-genome work done to date indicates that the pan-genome is larger than the genome of any individual, our work goes well beyond previous studies because the quality of the underlying assemblies and the controls and cross checks we employed allow us to have much higher confidence in the conclusions (genome assembly quality below). Interestingly, a comment for reviewer 4 supports the novelty of our work: "While the pan-genome has been an issue that has been well addressed in bacterial genomes thus far (with few exceptions) this hasn't been a target for plant genomes in general."

We agree that any manuscript can be improved by additional analyses and comparisons, but we feel that a comparative analysis between pan-genomes is an enormous undertaking that is well beyond the scope of the current project. Making meaningful comparisons between pan-genomes is a fascinating topic, but would require considerable time and effort and the full conclusions of such a study would be worthy of its own high-impact publication.

I am not sure what additional functional studies the reviewer has in mind, but two examples already in the manuscript illustrate the potential functional significance and comparative value of the pan-genome:

- 1) Correlation between variants associated with flowering time and population structure/flowering time including the absence of a gene, *Brdisv1ABR21022861m*, whose orthologs in other plants are involved in flowering time, from the early flowering populations.
- 2) The observation that pan-gene, *Brdisv1Bd1-11011965m*, is syntenic to a gene in wheat that is part of a locus known to harbour extensive PAV that affects disease resistance. These examples show that our results will allow scientists to develop and test hypothesis about specific pan-genes that they could not have developed using just the reference genome.

To expand our functional analysis, we added additional analysis of the interplay between population structure and pan-gene distribution to Fig 4 and greatly expanded the corresponding sections in the manuscript. This is novel for a de novo assembly-based approach because prior studies did not have sufficient numbers of individuals to examine population genetics.

The title is rather vague and perhaps misleading, I would recommend naming the species. The link to phenotypic variation is rather speculative and requires further support.

The title has been changed to:

“Extensive gene content variation in the *Brachypodium distachyon* pan-genome correlates with phenotypic variation”

We welcome other suggestions.

I agree that the small genome size makes it a relatively easy model for pangenome analysis, however the authors should consider how well findings translate to species with larger genomes, specifically those with greater TE content. One of the weak points of this manuscript is the lack of comparison with other pangenome studies. While the analysis of the *Brachypodium* data is mostly strong, without putting these findings into a broader context limits their interpretation.

As mentioned above, the type of meta-pan-genome analysis the reviewer requests is a huge undertaking. A new field really.

The main area which could be improved is in the validation of the assemblies and their annotation. Variation in assembly quality and annotation can lead to incorrect predictions of gene presence/absence variation and additional evidence supporting the gene variability presented here is required before conclusions can be confidently made. The authors acknowledge that 5 assemblies are of low quality and were removed from the analysis, but the quality of the other new assemblies is variable and is likely to impact on interpretation of gene loss.

Genome assembly quality

The quality and completeness of the assemblies is unprecedented for a pan-genome study. Figure 1a shows that all the assemblies contain nearly the same amount of coding and non-coding sequence as the reference genome with the exception of the few assemblies noted as being of lower quality. Even in these cases, it is mostly the non-coding DNA that is missing. The BUSCO scores for our individual genome support their completeness. Indeed, our best *de novo* assembly has a higher BUSCO completeness (Bd18-1: 98.4%) than the version 2 *B. distachyon* reference genome sequence (98.3%), which in turn is among the top ten most complete reference genomes currently on Phytozome (a large database of high quality reference genomes). The average completeness of all assemblies is 95.2% as estimated by BUSCO, which makes our average assembly nearly as complete as the rice reference genome (*Oryza sativa*, MSU v7.0: 95.6%). Only five of our assemblies have a completeness score less than BUSCO 90% (supp. table 2). All of our assemblies have far greater completeness as compared to the recently published reference genome for *Oropetium thomaeum* (70.2%), which is a small low-repeat grass genome similar to *B. distachyon* (VanBuren, Bryant et al. 2015). Thus the quality of the assemblies and annotations in this study is unparalleled.

Taken together I think that our data indicate that we have adequately addressed the issue of assembly quality and completeness. Furthermore, by confining our conclusions to comparisons between the 'shell' and 'core' pan-genes we avoid bias by artefactual annotations in the 'cloud' and missed annotations that result in placement in the 'soft core'.

Nevertheless, to highlight the consistency of all assemblies in number of primary transcripts and their categorization into cloud, shell, softcore, and core we added an additional panel figure 2b.

Verification of PAV

We verified the PAV using read mapping at two different scales. At the scale of individual loci we 1) mapped reads from multiple lines to a single assembly (genomic) as shown in figure 1d. It is clearly evident that the lines that lacked a particular shell gene do not contain any reads that align to that genomic segment whereas lines that do contain that gene have continuous read coverage. 2) for a larger deletion we mapped reads to an assembly containing the region from a lines with contrasting PAV. In both cases, the reference genome is missing the sequence indicating that we have captured sequences not contained in the reference genome.

To extend this type of analysis across the entire pan-genome, in figure 2d we mapped reads from each individual line to the CDS from all pan-genes (each pan-gene cluster is represented by a single gene from the pan-gene cluster). The percentage of each gene covered by short reads is represented by color-coding. As is evident from the picture, the core genes are covered essentially 100% in all lines and the shell genes lack coverage from many lines as predicted from our gene-based pan-genome. When the lines are clustered using the read coverage percentage, the tree is essentially identical to the SNP-based phylogeny in figure 4. Furthermore, supplemental figure 4 shows a similar clustering based on pan-gene CNV which, again, produces essentially

the same tree produced by the SNP-based analysis. This indicates that the observed PAV and CNV is consistent with the underlying biology and not due to random assembly artefacts.

Further supporting the accuracy of our PAV calls in supplementary figure 1c we plotted the number of non-reference genes contained in 3 of our assemblies that were not covered by reads (<80% of gene covered by reads) from the three lines and the line corresponding to the reference genome. In each case, the absence in the reference genome and presence in the new assemblies was confirmed and the amount of PAV in each genome was comparable.

To address the reviewer's concern that a few lines may disproportionately inflate the pan-genome, we plotted the number of core, soft core, shell and cloud gene in each line (Fig. 2b). In addition, we plotted the number of high-quality non-reference genes whose transcripts did not map to the reference genome in Fig. S1d. There are no outliers on either graph indicating that our estimate of pan-genome size was not adversely affected by poor assembly quality of individual lines.

On page 3, it is suggested that some genes and control elements are truncated. It would be valuable to quantify these and discuss in relation to genome assembly quality.

This comment is concerning our mention of a limitation of the way we constructed the sequence-based pan-genome. We constructed the sequence-based pan-genome purely as a control to reassure ourselves and readers that the pan-genome is much larger than the reference genome. By stopping the addition of sequences to the pan-genome when a stretch of 21 nucleotides already contained in the pan-genome is encountered we undoubtedly truncated genes. However, little biological insight would be gained by characterizing these arbitrary truncations. All we want readers to take away from this analysis is that the sequence-based pan-genome is similar in magnitude to the gene-based pan-genome. We altered the text in the manuscript to clarify this. The section now reads:

“To obtain a preliminary estimate of pan-genome size, purely at the DNA sequence level, we constructed a sequence-based pan-genome by iteratively scanning each of the 54 genome assemblies and adding DNA sequences that were > 600 bp (long enough to contain a gene) and did not contain any 21 bp sequence found in the preceding sequences (Supplementary Information). The sequence-based pan-genome was 430 Mb, 58% larger than the 272 Mb reference genome, and contained 40% more genes. The average length of the DNA segments added to the pan-genome was 1,487bp, much larger than the 600bp minimum length cutoff. These analyses reveal a large amount of gene and non-coding sequence that is not captured by a single reference genome.”

A very small number of genes were found in all lines. This contrasts with what is observed in other pangenome studies and suggests that some assemblies may be of poor quality. More details of gene loss across the lines are required – for

example, are some lines missing many more genes than others. Was there any relationship between 'lost' genes and repeat motifs?

We observed that 13,408 pan-genes were shared by all lines and that 7,823 pan-genes were contained in almost all lines (the 'soft core' category is included to acknowledge the imperfections of assembly and annotation). These are large numbers and consistent with the trends seen in other studies. While pan-transcriptome, meta-genome, or reference-based studies have low power to accurately identify new sequence and underestimate pan-genome size, even these studies are consistent with our results. For example, a study that generated over 32 Gb of sequence in the low-copy region of the genome across 27 diverse maize lines estimated that the B73 genome (reference genome) contained only 70% of the low-copy sequence in the maize pan-genome (Gore et al., 2009). Similarly, while pan-genome studies based on small numbers of de-novo assemblies individuals have little power to accurately estimate pan-genome size, as study of 7 *Glycine soja* accessions is consistent with our findings. "Approximately 80% of the pan-genome was present in all seven accessions". However, since the study only included 7 individuals the overall the pan-genome was underestimated. As shown in our simulation in supplementary Fig. 2a, a pan-genome derived from only 7 individuals is only 62% of the size estimated from all the lines in our study.

To clarify these areas we added an additional panel, Fig. 2b. The plot shows that the assemblies have roughly the same number of genes with similar numbers of genes in the four respective pan-genome categories. Slight exceptions to this are two of the lines, already mentioned the main text as lower performing assemblies. However, we do not observe an increase in cloud genes in those lines, indicating that their inclusion in our study did not inflate our estimates of pan-genome size. In fact, the line with the greatest number of cloud genes is not associated with one of the low performing assemblies. We also added the following text to the manuscript: "On average, any individual line in our study is composed of mostly core or softcore pan-genes (73%) and only 27% is categorized as shell or cloud (Fig. 2b). Thus, the majority of genes within any individual are found in all (or almost all) other individuals."

See below for analysis of genes with repeat motifs.

The terminology of core, soft-core, cloud etc. is new and seems arbitrary. Without some biological basis for this classification, I suggest using the more standard core and variable gene terminology.

There is no standard nomenclature. We adopted the nomenclature from this publication: Koonin, E. V. & Wolf, Y. I. Genomics of bacteria and archaea: the emerging dynamic view of the prokaryotic world. *Nucleic Acids Res* 36, 6688-6719 (2008). It is cited at the sentence in which the terminology is introduced. We used a 4 category system because it provides some information about the uncertainty of the categories where a gene is found in only a few lines or when a gene is only missing from a few lines. Using a binary system as the reviewer

suggests does not capture this uncertainty and would overestimate the significance of the findings. In addition, Reviewer 2 requested us to increase the number of categories and reviewer 4 was very satisfied with our classification. Thus, we kept our nomenclature.

The cloud genes were removed as likely artefacts, but some were then identified as real and expressed. Presumably it is possible to model the pangenome expansion with increasing number of lines to predict the true number of genes which are relatively rare in a representative population.

We did model the growth of the pan-genome in supplementary figure 2a.

The shell pan genes which are missing in some lines but not others require further validation such as PCR to ensure they are not artefacts.

This was addressed exhaustively by read mapping as described in the PAV validation section above. Testing a handful of loci with PCR will not increase the overall confidence because one could argue that a negative result is due to polymorphism in the primer sequence etc.

Genes with repetitive motifs tend to collapse in short read assemblies. It would be valuable to assess whether the variable genes had a greater frequency of repeat motifs. This is not clear from the current high level annotation presented.

The link between variable genes and disease resistance is highly speculative and needs much stronger support to be included. Disease resistance genes contain repeat motifs and while many do show presence/absence variation, their loss is also likely to be an artefact of assembly or annotation.

To address the concern that our assemblies may not accurately capture genes with repetitive domains we added Figures S1e-g. These show that our assemblies performed remarkably well at capturing genes (leucine rich repeat receptor-like protein kinases and NB-ARC genes, both often involved in disease resistance) that are difficult to assemble. All the lines (including the reference genome assembly) have a similar number of genes in these categories. We also show results for CNV based on our pan-gene clusters for 119 previously manually identified NBS-LRR genes (*Brachypodium distachyon* reference genome paper). The reference annotation and reference control (our annotated short read assembly) are remarkably similar in their copy number for these NBS-LRR genes. Thus, our pan-genome estimates should be robust even for genes with repetitive elements.

To clarify this in the manuscript we added the following text:

“Inspection of respective annotations revealed similar numbers of specific annotations, even for repetitive genes that are typically difficult to assemble (Supplementary Fig. 1f,g).”

And:

“On average, any individual line in our study is composed of mostly core or softcore pan-genes (73%) and only 27% is categorized as shell or cloud (Fig. 2b). Thus, the majority of genes within any individual are found in all (or almost all) other individuals. Ninety-one percent of known reference NBS-LRRs, which are notoriously difficult to assemble, had identical copy number between the assembly control and the reference (Supplementary Fig. 1e). Fifty-five percent of differences between the assembly control and the reference were associated with multi-copy pan-gene clusters differing by a single copy. Only a single (1/119) false PAV event was detected for a single copy reference NBS-LRRs cluster, which was not detected in the assembly control.”

The location of variable genes on pseudomolecules required detailed comparison with observations in other species.

We added the following text to the discussion to compare to what is known from soy:

“Our observations are consistent with previous results in *Glycine soja* where they noted a higher level of PAV in pericentromeric regions⁸.”

There is another recent report from maize (Swanson-Wagner et.al. *Genome Research* <http://www.genome.org/cgi/doi/10.1101/gr.109165.110>) that reports PAV elevated at the ends of chromosomes. However, they used a gene array-based approach and the majority of genes are on the distal ends. They did not present ratios or have de-novo assemblies so they did not really sample pericentromeric regions. Thus, we did not mention this paper. If the reviewer knows of other papers please let us know.

The wording of the association of variable genes with TEs is misleading, while I do not disagree with the observation that they are correlated, there is no evidence for one ‘influencing’ the other. TE movement may well be associated with gene movement and loss, but this is not demonstrated in this manuscript. The analysis of TE elements in relation to PAV is interesting, however additional details would make it much stronger. Could we find out more about types of TE elements found in vicinity of variable genes? Are some of the more likely to mediate PAV than others?

Class I elements were proportionately overrepresented upstream of shell genes. This would suggest that illegitimate recombination mediated by LTR retrotransposons could be a mechanism by which shell genes are lost/gained. It makes sense that class I TEs are lead to PAV since they remove DNA from the genome, versus class II, which would simply cut and paste it elsewhere in the genome. Class II should lead to CNV. In definition of the pan-genome we allow for gene movement and are therefore less sensitive to cut and paste events. In any case, we simply note the co-localization and postulate on mechanisms. We do not state that there is a definite role for TEs in movement/creation of shell genes.

We changed the discussion of this:

“The higher relative abundance of shell genes in non-syntenic blocks of the genome and the higher level intra-species TE insertions and deletions near

shell genes suggests TE dynamics as an important mechanism for shell gene creation and removal similar to the role of TEs in generating inter-species differences in gene content (Freeling, Lyons et al. 2008, Choulet, Wicker et al. 2010). In light of our study this concept may be expanded to intra-species variation as previously speculated (Woodhouse, Schnable et al. 2010). Our observations are consistent with previous results in *Glycine soja* where they noted a higher level of PAV in pericentromeric regions (Li, Zhou et al. 2014). Class I elements were proportionately overrepresented upstream of shell genes. This would be compatible with retrotransposon-mediated long terminal repeat (LTR)–LTR illegitimate recombination as a mechanism by which shell genes are lost/gained. Indeed, it has been suggested that LTR recombination actively counters retroelement expansion in *B. distachyon* and may partly explain its relatively low complement of repeats and small genome size (International Brachypodium 2010). In contrast, it has been suggested that retroelements persist for very long periods of time in the closely related Triticeae, which have high repeat content and large genomes. High repeat content complicates genome assembly and thus precludes non-reference-based pan-genome analysis. Nonetheless, advances in technology may soon enable such analyses in large genomes with higher repeat content such as maize, which is believed to have a large pan-genome of unknown size (Gore, Chia et al. 2009).“

It is understood that wild germplasm maintain traits which may be of agronomic value so this suggestion is not new and several examples are published. The significant number of variable genes associated with telomere annotation requires an explanation, could this be an artefact due to repetitive motifs in these genes?

This is not an artifact due to repetitive motifs for the reasons discussed above. The GO term “telomere maintenance” involves the enrichment of annotations involving nucleic acid-binding/OB-fold-like proteins, PIF1 helicases, and conserved telomere maintenance component 1. Proteins with such annotations may be involved in telomere biology, but also may have roles in DNA replication and repair. The possible enrichment of telomeric proteins is intriguing, as such proteins show rapid evolutionary divergence and poor interspecies conservation. For example, the human CTC1 protein sequence shares only 69% identity with mouse, 30% with zebrafish and 14% with *Arabidopsis* (Linger and Price 2009). More importantly, it has been shown that gene duplication across species has created telomere protein paralogs with novel functions. While one paralog may be part of a conserved telomere protein complex and have the expected function, the other paralog may serve in a completely different aspect of telomere biology (Linger and Price 2009). Our data indicates that telomeric proteins may be diverse at the intra-species level.

To clarify this in the manuscript we added the following text:

“GO enrichments were also observed for “telomere maintenance”, associated with telomeric proteins known to show rapid evolutionary divergence and poor

interspecies conservation, although this enrichment did not pass our FDR threshold ($p < 0.05$; $FDR > 0.05$). Previous studies have shown that gene duplication has created telomere protein paralogs with novel functions (Linger and Price 2009).”

Two assemblers were used, was there any difference between gene content that could be associated with these different assembly processes? I am surprised that the 75bp reads from the HiSeq2000 produced such poor assemblies compared to the 100 bp reads. Is this more to do with quality of the older reads than the length?

It is a combination of read length and sequencing depth. The sequence data was filtered prior to assembly, so poor quality reads and sequencing artefacts were removed.

CEGMA is now replaced by BUSCO, though it would be valuable to include both analysis.

We ran BUSCO on the proteomes and replaced the CEGMA graph with a graph of BUSCO scores in supplemental fig. 1. We added BUSCO scores to supplementary table 2.

It looks like RNASeq was generated for 36 accessions, but not used in the annotation. Using RNASeq could improve annotation of accession-specific genes not found in Bd21, Bd21-3, Bd1-1.

We agree that RNA-seq would improve annotation. Unfortunately, the RNA-seq from the 36 accessions is 3' tag sequence for expression analysis and is not appropriate for annotation. Importantly, additional RNA-seq would likely increase the number of novel annotated genes and further increase the size of the pan-genome. Thus, it is not expected to affect our conclusions that the pan-genome is considerably larger than the genome of any individual line.

The authors perform population analysis of the accessions, however much more could be done for this analysis. The analysis splits accessions into three clades based on SNPs. A similar clade structure was recovered based on PAV analysis. Authors do perform analysis of flowering time traits. However any further analysis of PAVs in relation to population structure is lacking. A more in depth analysis would allow novel insights into evolution of undomesticated plant species. What are the functions clade specific genes? Can those be linked to geographical locations? Do the relationships recovered based on PAVs and SNPs mirror each other completely?

We demonstrated an example of the utility of this type of analysis by comparing the pan-genome, flowering time and population structure. We moved supplemental fig 5 to the main text to increase the focus on population analysis. We also added supplementary tables 5 and 6 to give readers access to the flowering times for all lines. We also agree that many more interesting stories are lurking in the data, but they are beyond the scope of this paper and will

undoubtedly be the focus of future papers. We also point out that the other reviewers praised the comprehensive nature of our analysis.

I am not sure what do the authors mean by this: The first clade contains lines from multiple geographic locations that almost all exhibit an extremely delayed flowering (EDF+) phenotype (Fig. 2c, 4b, Supplementary Fig. 4), indicating that flowering time is a major factor in the divergence of populations (Fig. 2c, Supplementary Table 3). Underscoring the relationship between flowering time and the pan-genome. Bayes factor (BF) tests for potential correlated evolution of flowering time traits (Fig. 4c) and molecular variants in genes of known flowering regulators (Fig. 4e) revealed strong associations (Supplementary Fig. 5, Supplementary Tables 3, Supplementary Information) What exactly is the proposed relationship. Could the authors elaborate and perhaps put into the context of what is currently known about the role and function of specific flowering time genes?

The reviewer is correct. We have clarified the issue and have rewritten the paragraph as follows: “The first clade contains lines from multiple geographic locations that almost all exhibit an extremely delayed flowering (EDF+) phenotype, whereas the second clade contains lines from eastern and western Mediterranean locations that do not show the EDF+ phenotype (Fig. 2c, 4b, Supplementary Fig. 4). Additionally, Bayes factor (BF) tests for potential correlated evolution of flowering time traits (Fig. 4c) and molecular variants in genes of known flowering regulators (Fig. 4e) revealed strong associations with, respectively, EDF+ and non-EDF+ clades (Supplementary Fig. 5, Supplementary Tables 3, Supplementary Information), These results suggest that flowering time is a major factor in the divergence of populations (Fig. 2c, Supplementary Table 3).”

Reviewer #2 (Remarks to the Author):

The authors find a surprising amount presence-absence variation (PAV) for genes in the plant genus *Brachypodium* and characterize how such variable genes differ as a class. The authors further argue that at least some these PA polymorphisms may contribute to functionally important phenotypic variation.

Overall, I think the work is sound and is certainly substantial (there is, in fact, too much data presented to even discuss it all in the main MS or for me to give a thorough review of it all).

It is significant in demonstrating that a single reference genome can give only an incomplete enumeration of the gene catalogue for a eukaryote, expanding the application of the “pangenome” from its more customary domain of microbes. Further, the findings raise the possibility that the variation in gene content among lines may underlie functionally important phenotypic variation (but see the caveat below).

Major comments

1. (Needs to be addressed by revision) The authors assert that the three groups of lines should be considered a single species, and thus share a single

pangenome, based on the fact that crossing in the laboratory is possible and admixed lines exist in nature (line 234). However, the latter are rare, and crossability between distinct grass species is common, including even intergeneric crosses.

The F_{ST} values in pairwise comparisons range from 0.42-0.63, which suggests considerable genetic isolation.

The 'species' designation is of course an arbitrary human construct and our personal interpretation is that the most highly diverged clade might indicate an incipient speciation event. However, we don't think it is quite an independent species yet because, while not extensive, we do see evidence of recent gene flow between the clades (13% of lines are admixed so it is not rare) as evidenced by the admixture observed in the structure analysis and by examination of the genome. Indeed, the reference genome contains numerous islands that are very similar to lines in the EDF clade. Since this is the earliest flowering line it highlights the fact that genetic material has moved in the recent past from the latest to the earliest flowering lines. The two most admixed lines suggest that there is a significant amount of mixture between groups in some areas.

With respect to F_{ST} , we agree that there is evidence of significant genetic isolation. However, for the reasons stated above, we still consider all the lines to belong to a single 'species'. In addition, the highly selfing nature of *B. distachyon* may increase F_{ST} since there is less opportunity for genetic intermixing and increased opportunity for advantageous genotypes to dominate in certain environments, though this may also be a mechanism to accelerate speciation. Interestingly, *B. distachyon* exhibits a very high recombination rates when crosses do occur. Whether this is adaptive is not known.

While this manuscript is not meant to be an examination of taxonomic relationships and how to define them, it does suggest that the pan-genome could be used in this regard. While I don't want to speculate in the manuscript, one could argue that this sort of analysis could be used to define a species and perhaps could be used to revisit the species groupings in some of the examples of interspecific/intergeneric crosses the reviewer is referring to. However, for our specific case I think it would be very difficult to argue that we have distinct species when all tested lines tested are inter-fertile and 7 of 54 (13%) lines show a significant amount of admixture.

To clarify this in the text we added:

"Despite strong population structure, 7 of 54 (13%) lines in our study show a significant degree of admixture including large stretches of DNA in the earliest flowering line Bd1 (also sources of the reference genome) that look very similar to EDF lines indicating gene flow between the latest and earliest flowering lines. Not surprisingly then, all three groups are fully inter-fertile in laboratory crosses. Thus, despite the high F_{ST} rates and genomic structure this group still appears to functionally behave as a single species."

The EDF+ group, in particular, flowers late, a strong pre-zygotic reproductive isolation barrier, and shows 100-150,000 fixed SNP differences, though admittedly the case for S+ and T+ being distinct is weaker (Suppl Fig 4). The exact taxonomic rank of these groups is not important, but it does suggest analyzing the PAV variation in a hierarchical manner, since patterns of presence and absence in a gene may not be the same in all subtaxa.

This raises some interesting questions:

- a. Is a gene classified as shell gene among all subtaxa core to one or two of them?
- b. What is the differentiation among the taxa in gene content?
- c. Is the ratio of pangenome to reference smaller when the pangenome is constructed for each subtaxon separately?

“differentiation among the taxa in gene content” is presented in Fig. 4i.

- c. Is the ratio of pangenome to reference smaller when the pangenome is constructed for each subtaxon separately?

We agree that this is an interesting area to explore. We constructed pan-genomes using only non-admixed lines within the three population groups, respectively. The results are presented in Supplementary fig 2b and the following text was added to the manuscript:

“Fixed-SNP differences and F_{ST} estimates were greatest between the EDF+ group and the other two groups (Fig. 4k, Supplementary Methods and Results). As expected, the EDF+ clade contributed the most non-reference gene clusters on a per line basis due the reference genome residing in the distant T+ group (Fig. 4k). We observed hundreds of genes that were present in only one of the population groups (ignoring admixed lines, Fig. 4l). In order to further explore the interplay between population structure and the pan-genome, we constructed pan-genomes using only non-admixed lines within the three population groups, respectively. As expected from our simulations (Supplementary Fig. 2a), the size of the population group pan-genomes decreased with sample size: T+=48,481 pan-genes among 21 lines; S+=38,739 pan-genes among 16 lines; EDF+=37,742 pan-genes among 7 lines. However the number of non-reference genes in each of the three experiments is still significant. For the EDF+ group we identify 2,868 non-reference pan-genes (relative to the EDF reference line BdTR8i), in T+ group we identify 6,746 non-reference pan-genes (relative to the non-admixed T+ reference BdTR3c), and in the S+ group we identify 4,790 non-reference pan-genes (relative to the S+ reference ABR4). In addition, we observe the same pattern as the full pan genome when we plot the number of genes vs the number of lines containing each pan-gene (Fig. 2a, Supplementary Fig. 2b). This indicates that the ratio of the various pan-gene compartments is not determined by population structure. Interestingly, we find that some shell genes in the full pan-genome are core within the sub-populations: 868 pan-genes for EDF+ population, 1,501 pan-genes for S+ population, and 265 pan-genes for the T+ population (Fig. 4n). Shell pan-genes

that were core to the EDF+ population were enriched in GO terms including “regulation of gene expression” ($p < 0.05$, $FDR < 0.05$), which was also observed for shell pan-genes core to the S+ population ($p < 0.05$, $FDR > 0.05$). Shell genes core to either the T+ population or S+ population had enrichment for “multicellular organism development” ($p < 0.05$, $FDR > 0.05$). Despite strong population structure, 7 of 54 (13%) lines in our study show a significant degree of admixture including large stretches of DNA in the earliest flowering line Bd1 (also sources of the reference genome) that look very similar to EDF lines indicating gene flow between the latest and earliest flowering lines. Not surprisingly then, all three groups are fully inter-fertile in laboratory crosses. Thus, despite the high F_{ST} rates and genomic structure this group still appears to functionally behave as a single species.”

We also added a plot in Fig 4n that shows how many shell genes in the combined pan-genome are core in one of the sub-groups.

2. Related to that, It would be helpful to have a clearer visualization of the matrix of presence-absence variation of all the genes across all the lines than the current Fig 2c. A hierarchical clustering of the genes (and possibly the lines) in Fig 2c based on similarity in presence-absence vectors might make the patterns more evident, and illuminate, for instance, if the shell genes are distributed evenly among the subtaxa.

Fig 2c (now fig 2d) is already logically ordered by the number of genomes in which each pan-gene is represented. Genes found in all lines are at the top and genes found in only 3 lines are at the bottom. On top of this we display the read coverage of that pan-gene by the sequence data sets for each inbred line. To cluster the genes within this plot would remove its initial purpose, which was to validate the core, softcore and shell categories by independent read mapping data.

We have included a clustering of copy number for each line, for all shell pan-genes (Fig S4a) and added this in place of the older panel Fig. S4b.

3. The findings regarding phenotypic effects for Brdisv1ABR41022793m and Brdisv1Bd1-11011965m are very interesting and highly suggestive, but are not conclusive in linking phenotypic variation to PAV. The assertion in the title strikes me as too strong. If the authors still wish to make this one of the primary take home messages of the paper, I would at least suggest moving Suppl Fig 5 into the main MS, beefing up the superficial description of these results in the main MS, and making the case for the overall conclusion, including caveats, more explicit in the Discussion.

To better represent the correlative nature of the pan-genome to phenotype link we changed the title to: “Extensive gene content variation in the *Brachypodium distachyon* pan-genome correlates with phenotypic variation”

We moved part of fig S 5 into the main manuscript and added supplemental figure 6 that covers more ancestral trait predictions. We added the following text to the manuscript:

“To explore the evolutionary history of the population groups we reconstructed the ancestral state for the 25 discrete flowering time traits and molecular variants in genes of known flowering regulators recorded in 53 of the sequenced lines (Fig. 4c-e, Supplementary Fig. 6). Changes were inferred to have occurred for most analyzed traits along the long branch leading from the most recent common ancestor (MRCA) of *B. distachyon* (node 1) to the MRCA of the EDF+ clade (node 2), which showed an ancestral pattern, congruent with that observed today in most of its descendant lines (e. g., extremely delayed flowering and distinct polymorphisms in vernalization and flowering genes; Fig. 4a, Supplementary Table 3). By contrast, changes were less pronounced in the short branches leading to the respective MRCAs of the T+-S+ clade (node 8) and the S+ subclade (node 35) (Fig. 4a; Supplementary Table 3). Additionally, Bayes factor (BF) tests revealed strong evidence ($BF > 5$) of correlation between flowering time traits (Fig. 4C) and molecular variants in genes of known flowering regulators (Fig. 4e) (Supplementary Fig. 5, Supplementary Tables 3, Supplementary Methods and Results). These results suggest that flowering time is a major factor in the divergence of populations (Fig. 4a-c,e-j Supplementary Fig. 6; Supplementary Table 3). Conversely, there was weak correlation between ($BF < 2$) collection latitude and flowering phenotype and polymorphisms in flowering and vernalization genes indicating that latitude is not a major factor driving intra-specific divergence of flowering time.”

4. The authors conservatively exclude “cloud” genes from the analysis, and distinguish the “soft core” from both the “shell” and “core” classes, both of which seem sensible. But the lack of a soft-shell seems a bit arbitrary given the distribution shown in Fig 2a – there is a sizable shoulder of genes present in 2, 3, 4, 5 lines, and hardly any found at intermediate frequencies. If the authors prefer not to include a soft-shell category in a reanalysis, I would at least like to see a justification for the arbitrary cutoffs in the Methods, and an explanation in the Discussion about how choosing different cutoffs/categories would affect the interpretation.

The nomenclature used was an attempt to capture some of the technical uncertainty. This was described in the response to reviewer #1. Please note that reviewer 1 wanted us to use a binary classification, the opposite of the request by reviewer 2 to further divide the classification. We kept our nomenclature and added the following text to explain the rationale.

“This classification system allows us to represent some of the technical uncertainty particularly for the soft-core where true core genes could be placed if they were missed in a few annotations and the cloud which could contain anomalous annotations from one or two genomes. Thus, the most robust comparison is between shell genes and core genes which are much less likely to contain artifacts.”

5. It would be helpful to the reader to have more in the (unusually brief) Discussion comparing the general pattern of PAV seen here to what's been seen, or may be seen, in other systems. For instance, our group has demonstrated that there is purifying selection against gene deletions that are variable within a natural plant population (e.g. doi:10.1093/gbe/evt199, not cited).

The following text was added to the discussion and the paper mentioned cited:

“This fits a scenario where new pan-genes are continually created and lost unless they are adaptive under some conditions. Selection against indels has been noted in other systems(Flagel, Willis et al. 2014).”

6. Why would a maximum likelihood tree that forces bifurcations and does not allow reticulations be appropriate for modeling the evolutionary history of these lines? This is not a critical problem for this MS, since the interpretation of tree is incidental to the main conclusions. But in general, where there is ongoing mating and recombination, a phylogenetic tree is not the correct model for explaining similarities among genotypes, and is potentially misleading in its interpretation. A simple hierarchical clustering method would be more appropriate (see also point 5).

The ML tree is used as an approximate average genealogical tree. While not perfect, it is a good approximation for highly selfing species with very low gene flow, as in *B. distachyon*. The high bootstrap values support this approach. To provide some additional validation we used an additional method to create a tree and added the following text:

“To investigate the possibility that incomplete lineage sorting and using a single concatenated SNP data set(Kubatko and Degnan 2007) introduced topological errors we also ran a phylogenetic analysis with a method, Singular Value Decomposition quartet (SVDq)(Chifman and Kubatko 2014), that uses coalescence modeling. This approach is especially useful in cases with ongoing gene flow. Since the topology of the SVDq tree was very similar to the ML tree (Supplementary Fig. 4a,b) we focused on the ML tree for further analysis.”

Minor

7. Include the estimates of F_{st} within the main MS rather than the Supplementary Methods.

This was added to the main MS:

“ F_{ST} estimates were 0.5579, 0.6277, and 0.4220 respectively for EDF|T+, EDF|S+, and S+|T+ comparisons indicating that the groups are strongly differentiated.”

8. The Supplementary Methods contain quite a number of results, so the name is a little misleading.

The results in the methods section concern method optimization, not the biology of the system. We could move them to the main text but that would increase the length. If the editor agrees, we will change the name to “Supplementary methods and results”.

9. I am not familiar with the term “base perfect” and don’t see an explanation in the cited reference (Ref12).

‘base-perfect’ was changed to ‘a finished genome except for the placement of some centromeric repeats’

10. Spell out “FPKM” upon 1st occurrence.

Done

11. Is the legend in Fig 5b all correct? The text implies that TE abundance, shell:core ratio and nonsyntenic:syntenic ratio are positively correlated. But the latter two appear to be correlated with “TE absence” here.

What is plotted is the number of TEs found in the reference genome that are absent in one of the other lines. This is a measure of TE movement. To clarify, the legend was modified and now reads:

“TEs absent (a measure of TE dynamics) in other *B. distachyon* lines plotted for 2.5Mbp non-overlapping windows along chromosome 4. **c**, Plot of intra-species TE insertions relative to the reference genome (a measure of TE dynamics) versus shell/core gene ratio. Plot of intra-species TE “absence” relative to the reference genome (a measure of intra-species TE dynamics).”

12. I’m not sure I see the trend in Fig 6f that the authors are reporting.

The size of the orange bar varies for the shell genes. This is a minor point.

13. Line 779: wouldn’t the minimum BLAST coverage be more informative than the mean?

The mean indicates that most hits cover a high percentage of most of the genes.

14. Line 812: define or provide a reference for “Cscore”.

Cscore is a protein BLASTP score ratio to MBH (mutual best hit) BLASTP score. It was already defined at ~ line 620.

Typos and such

15. Fig 6f: legend color for ‘shell’ is pink not orange

It is orange on my computer. We specifically selected colors that would be more accessible to colorblind people (<http://jfly.iam.u-tokyo.ac.jp/color/index.html>). We'll keep an eye on the proof

16. Line 628. Is Ref15 the correct citation?

Yes.

17. Line 640L "miss-annotated" -> "mis-annotated"

Done.

18. Line 652. "R2" -> "R^2"

Done.

19. Line 657 Provide figure numbers.

I don't understand this one.

20. Line 983. Capitalize "Gene Ontology".

done

Signed, Todd Vision

Reviewer #3 (Remarks to the Author):

The manuscript from Gordon et al "A plant pan-genome links extensive variation in gene content to phenotypic variation" deals with a large pangenomic approach in the wild grass *Brachypodium distachyon*.

The study is very interesting of high importance in those years of population genomics and GWAS, and provides a lot of clues that can explain unexpected results from reference-based approaches.

The experiment it self is well conducted, with a lot of supporting data and analyses. The paper is well written, with for me few corrections, mainly deeper informations needed rather than missing ones (see below).

My two main questions is on the origin of the genes from what the authors called the shell and cloud genomes and on their potential impact on speciation. In page 4 and 5, the tried to identify a potential origin through duplication or drift from reference genes, and succeed for some, but not all. I would like to known if authors, even for discussion purpose, have any idea of the origin of those non-duplicated gene, ie of the neogenesis in this specific case: horizontal transfer ? Neogenesis per se (as in papers from Long, such as Evolution of New Genes, 2015. Oxford Bibliographies or Kemkemer C, Long M (2014). New genes important for development. EMBO Rep doi: 10.1002/embr.201438787) ? This is a tremendous subject that may explain 'rapid' adaptation of organisms to a wide range of environments.

The 'new genes' may simply be due to deletion of a gene that was in the last common ancestor and so may not be really new genes. We agree that this is a very interesting area to explore but feel it is beyond the scope of this already very large manuscript.

Class I TE elements were proportionately overrepresented upstream of shell genes. It is possible that illegitimate recombination mediated by LTR

retrotransposons could be a mechanism by which shell genes are lost/gained. This contrasts with class II TE elements which cut sequence and paste it else where in the genome. Cut and paste activity is less likely to lead to PAV. In defining the pan-genome we allow for gene movement and are therefore less focused on cut and paste events that do not lead to PAV.

We agree and added the following text to the discussion:

“Class I elements were proportionately overrepresented upstream of shell genes. This would be compatible with retrotransposon-mediated long terminal repeat (LTR)–LTR illegitimate recombination as a mechanism by which shell genes are lost/gained. Indeed, it has been suggested that LTR recombination actively counters retroelement expansion in *B. distachyon* and may partly explain its relatively low complement of repeats and small genome size(International Brachypodium 2010).”

In the same way, speciation is known to be at least partially linked on recombination issue in zygote formation or in meiosis errors in hybrids. Thus, is there any clue about the impact of such pan-gene difference on the recombination/meiosis, such as difficulties to obtain a 100% fertile descent in crosses between individual having highly different dispensable/shell genome ? It could be interesting to discuss this issue, but perhaps it is a too big one for the current scope of the paper.

We agree that this is an interesting area to explore. It is simply beyond the scope of this manuscript. To do this correctly would require extensive additional experiments and characterization of genetic crosses.

As other requests/corrections/point of discussions:

1- Why authors use the terms shell and cloud genomes instead of the generally used dispensable and individual ones ?

As mentioned above:

There is no standard nomenclature. We adopted the nomenclature from this publication: Koonin, E. V. & Wolf, Y. I. Genomics of bacteria and archaea: the emerging dynamic view of the prokaryotic world. *Nucleic Acids Res* 36, 6688-6719 (2008). It is cited at the sentence in which the terminology is introduced. We used a 4 category system because it provides some information about the uncertainty of the categories where a gene is found in only a few lines or when a gene is only missing from a few lines. Using a binary system as the reviewer suggests does not capture this uncertainty and would overestimate the significance of the findings. In addition, Reviewer 2 requested us to increase the number of categories and reviewer 4 was very satisfied with our classification. Thus, we kept our nomenclature.

2- p3, l 118-119: what is the median length of added segments ?

The median is 1040, close to the mean presented 1,487bp

3- p4, l140: how did you limit the soft-core/shell frontier ?

The cutoff was conservative to represent the number of times we thought we might miss an annotation.

4- p4, l144-145: not sure that most cloud gene are artifact... Did you check them for Pfam or other system thoroughly for verifying their potentiality ?

We are not arguing that most cloud genes are artifacts just that they are likely enriched for artifacts so we excluded them from some analyses to increase our confidence in the results.

5- p4 l148-149: 22% of them are expressed in leaves (so many more in the whole life of the plant). What is the relative part of core genes expressed in leaves ? This could be an indication of the true number of real cloud genes (individual specific ?)

Percent of pan-genes expressed:

22% of cloud genes

96% of core genes

70% of shell genes

6- p4 l160-166 and latter on: all your synteny outside the Brachypodium are based on one, max 2, genome per species, and thus may be also biased in terms of conclusions. i would have been less strong in conclusions about their relatives in other species. Perhaps they exist but we did not find them yet. This could be tested e.g. on rice using the RPAN data (<http://cgm.sjtu.edu.cn/3kricedb/>), but it is not mandatory here.

We agree that there may be differences when we look at additional individuals. However, the difference will probably be small, since the shell genes are less conserved, and a more thorough analysis would be an interesting paper in itself. That would be a move toward a new pan-genome era.

7- p6 l238: where are the copy number (CNV ?) analyses ? I found PAV, SNP but not CNV there. Mistypo ?

A plot of CNV is shown in Fig. S5a and it reassuringly shows the same trend as the PAV and SNP.

8- p7 and generally: are they some genes that are never associated (ie antagonist?)?

This is an interesting idea but we did not do this analysis.

9- p8 For TE analysis, is there any link with the TE size and nature/level of expression of genes ? I mean if TE are linked to shell genes, normally they would be more recent copies, thus longer than background ones ?

This is an interesting question. However, we used breakpoints of homology between paired-reads to map TE movements. We believe that this approach overcome the potential problem of TE mis-assembly. TEs are by definition more difficult to assemble than genes with short reads due their high copy numbers. Therefore we have little information about the length of the respective elements. However, a manual inspection of the reads overlapping newly inserted elements (100 elements) and their flanking sequences indicates that we are most likely dealing with full-length elements. Indeed, such reads overlap for the large majority with the exact beginning and end of the consensus sequence of the element. Some of these insertions may correspond to solo-LTR. Yet, the fact that new element insertions occur at low frequency across the 53 genomes is an indication that TE activity was recent and we believe that most newly inserted copies are full-length.

10- p8 core genes, if they are essential, will be expected to be more transcribed than shell one...

We agree with the reviewer and this is what was shown in fig. 3e.

Supp Methods:

11- in general, an effort must be done there to homogeneize the writing. Please correct the version of software (sometimes given, sometimes not), the abbreviations (given sometimes after having used them), the text in itself (p17, l520-521 text is already written above).

I prefer all versioning in the methods or supplement.

Versions were added and standardized in the methods.

12- p17 l530-531: 5 to 25% of sequences cannot be mapped (234kp relative to 1Mb genes): I suspect it is mitochondrial/chloroplastic data ?

It is only 0.1% not mapped (234kb vs 272Mb genome). In addition, we filtered organellar DNA.

13- p19 l607: why having worked on 27 lines only ? Are they representative of diversity ?

We only had RNA-seq data for 36 lines and 9 of those were excluded from the analysis because the assemblies were lower quality and we were looking at non-coding sequence. We mentioned that poor assemblies were excluded from some analyses.

14- p19 l648: please provide the correct link.

Done. (<http://botserv2.uzh.ch/kelldata/trep-db/index.html>).

15- p20 l668: it finished quite abruptly...

The discussion was modified and expanded a bit but we are constrained by word count.

16- p20 l684-686: the selection is based on alphabetical order and thus will be biased in direction of a certain type of diversity ?

Pan-genes belonging to the same cluster are, by definition, very similar. Thus, while there may be some differences due to the representative picked, it should be minimal. The internal consistency of our results and successful cross validation indicate that there are no large systematic errors.

17- p22 Pangenome size simulation: can you estimate the minimal number of genomes to be sequenced to close it with your data ? It is not really clear...

To clarify this the following text was added:

“To examine how many lines need to be sampled to capture the *B. distachyon* pan-genome we conducted simulations of pan-genome size from increasing numbers of randomly selected lines. The results indicate that the pan-genome increases rapidly up to 20 lines and is still steadily increasing at 54 lines (Supplementary Fig. 2a). Thus, even 54 lines does not capture the full diversity of the species.”

18- Supp figures 6k-0 are completely unreadable...

Dashed lines were replaced with solid lines to increase readability.

In conclusion, I think the paper is of high interest and must be published as soon as those questions and corrections will be addressed

Francois Sabot

Reviewer #4 (Remarks to the Author):

Gordon et al., A plant pan-genome links extensive variation in gene content to phenotypic variation

The authors embark towards sequencing and comparative analysis of 54 different cultivars of *Brachypodium distachyon*, an important model organism for grasses (in particular cereal grasses) in general. The manuscript is well structured and very appealing to read and to a large extent I've been impressed by the clear and straightforward analysis, the very good documentation in the supplementary material and the nice and illustrative figures.

While the pan-genome has been an issue that has been well addressed in bacterial genomes thus far (and with few exceptions) this hasn't been a target for plant genomes in general. Economics of genome sequencing and resequencing now allow to address this question also in larger eucaryotic genomes. The

authors used a very nice and illustrative classification schema and dissected the genome(s) into core, shell and cloud constituents with the conclusion that lots of additional sequences and genes are found (or not found) in different cultivars. Structure analysis subsequently subdivides the different cultivars into early flowering and cultivars that stem from Spain and Turkey. Some criticism here: Looking at figure 4f and the localities of where the cultivars have been collected and the STRUCTURE analysis this becomes somewhat trivial. Collection areas are geographically separated anyhow.... However the analysis is to a large extent very robust and well documented and I can envision that future pangenomic analysis in various plant genomes might use the Brachypodium approach as a blueprint, a motivation and an inspirational resource

Criticism:

only few but...

line 116: "...and did not contain any 21 bp sequence found..." This is really enigmatic and even from crosschecking the supplementary material it doesn't become very clear. Well since this reviewer has some background in bioinformatics and genome assemblies I can imagine what this means but for the less expert readers it might help to render this a bit more understandable

To clarify this we changed the description to:

"To obtain a preliminary estimate of pan-genome size, purely at the DNA sequence level, we constructed a sequence-based pan-genome by iteratively comparing each of the 54 genome assemblies to the preceding pan-genome to identify novel sequences > 600 bp (long enough to contain a gene). We defined sequence as novel if it did not contain a single 21 bp sequence found in the preceding pan-genome (Supplementary Methods and Results)."

- I am not clear on whether any efforts were undertaken to filter for potential contaminants. It would be easy to increase a pan-genome by all kinds of bug sequences....

Most DNA samples were extracted from purified nuclei preparations, thus lowering the amount of microbe contamination. Furthermore the libraries are unamplified also reducing the likelihood of preferentially amplifying contaminants. The resulting sequence was then filtered for known contaminants, including human and common microbe contaminants prior to genome assembly. These measures reduce the likelihood of including non-plant derived sequences in our assembled genomes. Furthermore, our analysis mainly focuses on annotated genes and our gene annotation pipeline either requires homology or expression support for a model to be accepted as valid. Eukaryotic (mainly plant) peptides were used for homology search and the RNA-Seq data used for gene model support was polyA-selected (thus mainly eukaryotic). Therefore, even if prokaryotic sequences were in the assembly, it is unlikely that they would pass our annotation filters and be promoted as valid gene models. Lastly, we performed BLAST against databases including the NCBI nr database. Focusing on non-reference genes, which are putatively more questionable than the reference sequences, we find that out of 3,129 clusters with matches, 3,085 are plant sequences (98.5%), and <10 sequences

are potentially microbial, which is <0.3%.

- what kind of genes are in the shell and cloud category? Did I miss this?

We excluded cloud genes from most characterization because we want to be conservative. That said, many cloud genes match known genes but they are enriched for genes with no homology to known genes consistent with an enrichment in artifacts.

-numbers: the authors are talking about 61155 pan genome clusters. A striking number that doesn't seem to fit to the numbers you give in figure 4i and would translate in a massive inflation of genes given that the reference genome has "only" 31000 genes or so. Please clarify

The 61,155 pan-gene clusters include the cloud subset. As mentioned in the main text, we do not focus on the cloud subset as it may contain a higher rate of artefacts, but rather focus on the higher confidence set of 37,866 pan-genes left after removing the cloud subset. Subsequent analysis in the paper focuses on these 37,866 pan-genes, including Figure 4m. Figure 4m had less than 37,866 pan-genes, as some admixed lines were removed in order to better focus on genes that are more prevalent in one of the subpopulations. We modified Figure 4m to include those admixed lines so that the total pan-gene number 37,866.

We also show a new Venn diagram to show how shell genes in the combined pan-genome are core to one or more of the structure population groups (Fig. 4n). In this figure we omit admixed lines since they are outliers in terms of their complement of pan-genes that are core to the sub-populations.

- line 178/179: 80% sequence similarity is used as a cut-off to distinguish or speculate about functional divergence. Where does this value come from? It certainly depends on how deep you go in the functional conservation and regulation interferes as well... well a broad field.

It has been observed in other studies that roughly below 80% enzymes start to diverge in function (see figure 1 of "Practical Limits of Function Prediction" (Devos and Valencia 2000)).

- line 199 ff and figure 3d and 3e: is there a measure of statistical significance

Yes. The p-values were added to the legend.

- line 201: "shell genes are also are less likely..." remove one "are"

Done

- line 203: "...core genes are expressed at higher levels and are more broadly expressed..." How is broadly defined? I understand as a biologist but I wonder about the definition used in mathematical terms.

This is stated in the methods: Narrow 3 or less tissues with FPKM greater than 3. Broad is defined as 7 or greater tissues with FPKM greater than 3. This uses the Davidson FPKM data for the reference line.

- line 236: To be honest I don't get the admixture argument for Arn1 and Mon3. Either my printout is of bad quality or it looks (in tendency) very similar to the neighbouring cultivars (fig 4b)

I think maybe the printout was not clear. There is clearly >10% of the genome that looks like the S+ group.

- line 363 ff: "...some shell genes in crop plants may encode traits of agronomic value..." Well maybe yes, maybe no.... After all Brachypodium is not a crop plant and even if this is "only" a speculation in the discussion I don't see a value in this. Leave out (?)

We prefer to leave this in since it highlights the potential value in studying pan-genomes. It is well documented that there is extensive PAV in resistance genes and that many of those are agronomically important. Thus, this statement is reasonable.

-the BrachyPan website and availability of data: when I checked the website it was "temporarily unavailable". Please fix and make sure the data become publically available (along with a potential publication). I have to say that so far my experiences with JGI and JGI data release policy was always very positive but I've seen very bad examples as well. Certainly not wishworthy

It is working. Maybe the reviewer checked during a network outage.

References

- Chifman, J. and L. Kubatko (2014). "Quartet inference from SNP data under the coalescent model." *Bioinformatics* **30**(23): 3317-3324.
- Choulet, F., T. Wicker, C. Rustenholz, E. Paux, J. Salse, P. Leroy, S. Schlub, M. C. le Paslier, G. Magdelenat, C. Gonthier, A. Couloux, H. Budak, J. Breen, M. Pumphrey, S. Liu, X. Kong, J. Jia, M. Gut, D. Brunel, J. A. Anderson, B. S. Gill, R. Appels, B. Keller and C. Feuillet (2010). "Megabase level sequencing reveals contrasted organization and evolution patterns of the wheat gene and transposable element spaces." *Plant Cell* **22**(6): 1686-1701.
- Devos, D. and A. Valencia (2000). "Practical limits of function prediction." *Proteins* **41**(1): 98-107.
- Flagel, L. E., J. H. Willis and T. J. Vision (2014). "The standing pool of genomic structural variation in a natural population of *Mimulus guttatus*." *Genome Biol Evol* **6**(1): 53-64.
- Freeling, M., E. Lyons, B. Pedersen, M. Alam, R. Ming and D. Lisch (2008). "Many or most genes in Arabidopsis transposed after the origin of the order Brassicales." *Genome Res* **18**(12): 1924-1937.

Gore, M. A., J. M. Chia, R. J. Elshire, Q. Sun, E. S. Ersoz, B. L. Hurwitz, J. A. Peiffer, M. D. McMullen, G. S. Grills, J. Ross-Ibarra, D. H. Ware and E. S. Buckler (2009). "A first-generation haplotype map of maize." *Science* **326**(5956): 1115-1117.

International Brachypodium, I. (2010). "Genome sequencing and analysis of the model grass *Brachypodium distachyon*." *Nature* **463**(7282): 763-768.

Kubatko, L. S. and J. H. Degnan (2007). "Inconsistency of phylogenetic estimates from concatenated data under coalescence." *Syst Biol* **56**(1): 17-24.

Li, Y. H., G. Zhou, J. Ma, W. Jiang, L. G. Jin, Z. Zhang, Y. Guo, J. Zhang, Y. Sui, L. Zheng, S. S. Zhang, Q. Zuo, X. H. Shi, Y. F. Li, W. K. Zhang, Y. Hu, G. Kong, H. L. Hong, B. Tan, J. Song, Z. X. Liu, Y. Wang, H. Ruan, C. K. Yeung, J. Liu, H. Wang, L. J. Zhang, R. X. Guan, K. J. Wang, W. B. Li, S. Y. Chen, R. Z. Chang, Z. Jiang, S. A. Jackson, R. Li and L. J. Qiu (2014). "De novo assembly of soybean wild relatives for pan-genome analysis of diversity and agronomic traits." *Nat Biotechnol* **32**(10): 1045-1052.

Linger, B. R. and C. M. Price (2009). "Conservation of telomere protein complexes: shuffling through evolution." *Crit Rev Biochem Mol Biol* **44**(6): 434-446.

VanBuren, R., D. Bryant, P. P. Edger, H. Tang, D. Burgess, D. Challabathula, K. Spittle, R. Hall, J. Gu, E. Lyons, M. Freeling, D. Bartels, B. Ten Hallers, A. Hastie, T. P. Michael and T. C. Mockler (2015). "Single-molecule sequencing of the desiccation-tolerant grass *Oropetium thomaeum*." *Nature* **527**(7579): 508-511.

Woodhouse, M. R., J. C. Schnable, B. S. Pedersen, E. Lyons, D. Lisch, S. Subramaniam and M. Freeling (2010). "Following tetraploidy in maize, a short deletion mechanism removed genes preferentially from one of the two homologs." *PLoS Biol* **8**(6): e1000409.

Reviewers' comments:

Reviewer #1 (Remarks to the Author):

Irrespective of the number of individuals or the quality of assemblies, many of the conclusions presented here have been observed in other plant pangenome studies and not mentioning this in the text suggests that these findings are novel, which they are not. I am not requesting a meta pangenome analysis but rather for the results presented here be presented in the perspective of previous findings, something which is usual for scientific publications.

While it is an improvement to include the species in the title, highlighting the correlation with phenotypic variation is misleading. As also pointed out by another reviewer, this correlation is weak and would require more support for it to be a focus of the manuscript.

Regarding the quality of the assembly, having some assemblies of high quality does not mean that the pangenome is of high quality, when some assemblies are admittedly of low quality. Validation is required, especially when the finding that there are very few genes found in all lines contrasts with other plant pangenome studies. The boldest statements require the strongest support.

The terminology of core, soft-core, cloud etc. was also questioned by another reviewer. While this may have been used once in a microbial pangenome, I recommend that the authors read the numerous plant pangenome papers where they will see that the two level nomenclature is used in all of them. The authors on one hand claim to have excellent assemblies but then introduce classifications and nomenclature which highlights the uncertainty in the results.

Regarding the collapse of repetitive genes, the argument that some genes have assembled does not mean that all have. Even the best assemblies collapse some repetitive genes such as NRS LRRs, the important aspect is understanding how much collapsing there is and accounting for this in the interpretation of the results.

Reviewer #2 (Remarks to the Author):

The authors have satisfactorily responded to the extensive comments from the earlier submission. The paper overall is sound as far as I can judge, and the findings are important and sufficiently well-documented.

A couple points the authors may choose to address in the final version:

1. In the pgph starting line 315, is what the authors refer to as Figure 4 in the text actually Fig 5? (not sure as the main figures were not included in the revised MS)
2. The relationship of the two ASR phylogenies shown in Supp Fig 5a is not clear to me from the legend.

Reviewer #3 (Remarks to the Author):

The current version of the manuscript has been widely improved and is of great interest.

Thus I agree for it publication

Reviewer #4 (Remarks to the Author):

Almost all of my concerns and criticism has been adressed and those that have been not adressed by the authors are either very minor or the authors gave a reasonable response on why they prefer to not adress the issue.

I'd like to see the work and analysis published asap.

Klaus Mayer

Reviewer #1 (Remarks to the Author):

Irrespective of the number of individuals or the quality of assemblies, many of the conclusions presented here have been observed in other plant pangenome studies and not mentioning this in the text suggests that these findings are novel, which they are not. I am not requesting a meta pangenome analysis but rather for the results presented here be presented in the perspective of previous findings, something which is usual for scientific publications.

We mentioned previous relevant studies that were published at the time of submission, 9 months ago, in the introduction. However, we were rather brief due to space limitations. We greatly expanded the discussion of previous studies in the introduction and discussion sections. In addition, we added three papers that came out after our initial submission. Below are the modified introductory paragraph and the discussion.

Modified introductory paragraph:

Pan-genomes have been created for some bacterial species and, typically, are much larger than the genome of any individual strain¹. In contrast, the challenges associated with creating multiple high-quality eukaryotic *de novo* genome assemblies and associated sequence annotations have prevented large-scale, in-depth exploration of eukaryotic pan-genomes. Rather, eukaryotic pan-genome studies have employed several approaches to avoid the difficulty of generating many high-quality genome assemblies: using reference genome-based approaches with targeted *de-novo* assembly²⁻⁷, focusing on a small number of relatively low quality *de novo* assemblies^{8,9}; employing a metagenome approach that combines low depth sequences from many lines with targeted *de-novo* assembly¹⁰; or creating a pan-transcriptome as a way to reduce complexity^{11,12}. While these studies all have limited ability to capture and describe the full nuclear pan-genome, most suggest a pan-genome that is considerably larger than the genome of any individual line. For example, a study of the maize pan-transcriptome suggested that the reference genome only contained half the genes in the maize pan-genome¹¹, a study of the low-copy regions of 18 wheat lines found 21,653 predicted genes that were not contained in the reference genome despite the fact that the lines were closely related⁴, and a metagenome study of rice found 8,000 genes that were not in the nipponbare reference genome¹⁰. The rice study also performed a genome wide association study that showed a remarkable 41.6 % of trait-associated SNPs were from genomic locations corresponding to non-reference genes. Thus, plant pan-genomes are potentially large and a source of important traits.

New discussion:

The primary goal of our study was to accurately estimate the size of a plant pan-genome. As mentioned in the introduction, the challenges of producing numerous, complete *de novo* genome assemblies have prompted previous plant pan-genome studies to use approaches that avoid whole genome *de novo* assemblies (e.g. reference-based with targeted assemblies, pan-transcriptomics, metagenomics) or examine small numbers of lower quality assemblies. These studies likely

underestimate the size of the pan-genome because they have limited power to detect novel contiguous sequence outside the reference genome or they use a small number of highly fragmented assemblies of unknown completeness. Nevertheless, they provide important insights into functional aspects of plant pan-genomes. For example, reference-based approaches found that non-reference genes are often involved in processes related to traits of agronomic interest such as environmental stress and plant defense responses and may be implicated in heterosis^{4,5,11-13}. They have also shown that as much as 30 percent of genes in a reference genome may be affected by PAV^{5,6,14}. Studies focused on *de novo* assembly approaches have observed similar functional attributes of genes associated with PAV across the species-wide pan-genome^{9,10}. These studies have also shown that up to 16 percent of the species-wide pan-genome lies outside the reference genome⁸. In pairwise comparisons, maize inbred lines show extreme PAV and copy number differences, which if extrapolated would lead to an immense species-wide pan-genome for this species^{15,16} suggesting that maize pan-genome size estimates represent a lower bound on pan-genome size.

By including assembly controls and conducting extensive cross validation of PAVs via read mapping and phylogenetic analysis, our study provides strong evidence that the *B. distachyon* pan-genome is considerably larger than the genome of any individual plant of this species. Indeed, the high-confidence pan-genome contains 7,135 pan-genes that were not contained in the reference genome and nearly half of the genes in the high-confidence pan-genome are found in only a subset of lines. The completeness and number of *de novo* genome assemblies utilized in our study, the selection of lines to maximize the sampled genetic diversity, and the fact that *B. distachyon* is a wild plant that has not experienced a domestication bottleneck all contributed to the large pan-genome observed for this species. Our estimated size of the *B. distachyon* pan-genome is a conservative estimate due to collapsed tandem repeats, co-clustering of related paralogs within an assembly to the same pan-gene, and lack of gene annotations for novel genes lacking expression or homology support (required for our gene annotation). Thus, we have defined a lower bound for the size of the *B. distachyon* pan-genome. Precisely defining the core genome becomes more problematic as the number of genomes sampled increases due to the challenge of assembling and annotating a particular gene model correctly in every sequenced genome. Our soft-core category reflects this uncertainty. It should be noted however, that uncertainty about the exact set of core genes does not affect our estimate of pan-genome size, our primary objective, because the pan-genome is simply the sum of the genes in the reference genome and all the non-reference pan-genes. While the *B. distachyon* pan-genome is larger than the pan-genomes reported for several other species, it is not completely unexpected based on results from other species. For example, an estimation of the maize pan-genome based on the pan-transcriptome¹² was of similar magnitude to the *B. distachyon* pan-genome. In addition, a study of 10 reference-guided assemblies of *Brassica oleracea* found that 20% of pan-genes showed PAV⁵ which is consistent with our simulation of the pan-genome size for 10 *B. distachyon* lines (Supp. Fig. 2A). Similarly, true *de novo* assemblies of 15 *Medicago* genomes found that 42% of genomic sequence was found in only some accessions⁸.

Previous studies utilizing a purely *de novo* assembly strategy have focused on relatively small numbers of less complete assemblies making it difficult to ascertain the effect of population structure on pan-genome size and phylogenetic distribution. Powered by the much larger sample size in our study,

we show the importance of population structure in elaboration of the species pan-genome. In our study, PAV correlates with phylogenetic relatedness and pan-genes that are non-core in the species-wide pan-genome are often core within sub-populations. After excluding admixed lines, the three major population groups in *B. distachyon* differ greatly in their complement of pan-genes. Hundreds of pan-genes are core to one sub-population while not found in other populations (Fig. 4n). Sub-population core genes may in fact contribute to perpetuating population structure. For example, we identify a putative NF-YB transcription factor in all EDF+ lines that could potentially contribute to flowering phenotypes that could reinforce the genetic distinctness of the EDF+ sub-population. Genes that appear dispensable within the species-wide pan-genome may in fact be extremely important for the biology of a sub-population¹⁷. Individuals from previously non-sampled sub-populations contribute far more to increases in pan-genome size than the addition of closely related individuals. This underscores the importance of careful selection of individuals for pan-genome studies, particularly in species like *B. distachyon*, in which geography may be a secondary factor in shaping population structure, and autogamy may influence the spread of and selection on PAVs. ENREF_62 The degree of population structure within a species and genetic bottlenecks, such as domestication, need to be taken into account when interpreting pan-genomes and may be reasons some crop plants may have smaller pan-genomes than wild species such as *B. distachyon*. Despite the large overall size of the *B. distachyon* pan-genome, individuals in our study share the vast majority (90% on average) of their genes at the pairwise-level. Previous studies have observed greater amounts of pairwise PAV between individuals in other plant species^{15,16,18}. Thus, the amount of PAV that we observe in the compact genome of *B. distachyon* is likely to be dwarfed by PAV in larger, more complex genomes such as Maize¹⁵.

The functional enrichments observed in the shell genes, their expression levels and patterns, and their high evolutionary rates are consistent with a scenario in which shell genes evolve rapidly and are more likely than core genes to be adaptive under certain environmental conditions. For example, we found that shell pan-genes are enriched in RIPs, which can provide an advantage in the presence of pathogenic fungi or herbivorous insects. In contrast to higher frequency pan-genes in the shell compartment, the long tail in pan-gene frequency in Fig. 2a may indicate that a large portion of PAVs is at low frequency and may be under negative purifying selection. This fits a scenario where new pan-genes are continually created and lost unless they are adaptive under some conditions. Selection against indels has been noted in other systems¹⁸. The higher relative abundance of shell genes in non-syntenic blocks of the genome and the higher level of intra-species TE insertions and deletions near shell genes suggest TE dynamics as an important mechanism for shell gene creation and removal, similar to the role of TEs in generating inter-species differences in gene content^{19,20}. In light of our study, this concept may be expanded to intra-species variation as previously speculated²¹. Our observations are consistent with previous results in *Glycine soja*, where a higher level of PAVs in pericentromeric regions was noted⁹. Class I elements were proportionately overrepresented upstream of shell genes. This would be compatible with retrotransposon-mediated long terminal repeat (LTR)–LTR illegitimate recombination as a mechanism by which shell genes are lost/gained. Indeed, it has been suggested that LTR recombination actively counters retroelement expansion in *B. distachyon* and may partly explain its relatively low complement of repeats and small genome size²². In contrast, it has been proposed that retroelements persist for very long periods of time in the closely related Triticeae, which have high

repeat content and large genomes. High repeat content complicates genome assembly and thus precludes non-reference-based pan-genome analysis. Nonetheless, advances in technology may soon enable these analyses in large genomes with higher repeat content such as maize, which is believed to have a large pan-genome of unknown size³.

The pan-genome, individual genome assemblies and associated data sets as well as interactive tools for mining this data are available at the BrachyPan website (<https://brachypan.jgi.doe.gov/>). These tools, in combination with the experimental resources available for *B. distachyon*, allow further investigation of the mechanisms and functional consequences of intra-species gene dynamics.

While it is an improvement to include the species in the title, highlighting the correlation with phenotypic variation is misleading. As also pointed out by another reviewer, this correlation is weak and would require more support for it to be a focus of the manuscript.

We changed the title to “**Extensive gene content variation in the *Brachypodium distachyon* pan-genome correlates with population structure**” This is strongly supported by the fact that clustering the lines based on PAV, and copy number variations recovers the same population groupings identified by SNP markers and prior studies using SSR markers.

Regarding the quality of the assembly, having some assemblies of high quality does not mean that the pangenome is of high quality, when some assemblies are admittedly of low quality. Validation is required, especially when the finding that there are very few genes found in all lines contrasts with other plant pangenome studies. The boldest statements require the strongest support.

I will address this concern in 3 parts: First, our rationale for including all assemblies and how our approach ensures that our conclusions were not distorted by the few assemblies with lower quality. Second, the controls we presented to demonstrate the validity of the detected presence/absence variants and comparison to a new dataset for further validation. Third, the consistency of our results with previous pan-genome studies.

1. Rationale

The main goal of this project was to identify sequences/genes that are not found in the reference genome in order to estimate the size of the pan-genome of a wild plant. Thus, we felt it was important to include all of the assemblies in constructing the pan-genome because each one harbored considerable diversity. A few lower quality assemblies would only be an issue if they inflated the overall size of the pan-genome. We took steps to ensure and demonstrate that this was not a significant problem, including the exclusion of cloud genes (found in only one or two genomes) to create a ‘high-confidence pan-genome’. As mentioned in the manuscript many of the cloud genes are undoubtedly real so we are being conservative by design.

The other type error is missing genes in some genomes due to incomplete assemblies. This error will not increase the size of the pan-genome so it is not a concern for our primary goal. It will, however, affect the set of core genes found in every line. This is the rationale for inclusion the soft-core category as already mentioned in the results section. To clarify this we added the text below to the discussion:

“Precisely defining the core genome becomes more problematic as the number of genomes sampled increases due to the challenge of assembling and annotating a particular gene model correctly in every sequenced genome. Our soft-core category reflects this uncertainty. It should be noted however, that uncertainty about the exact set of core genes does not affect our estimate of pan-genome size, our primary objective, because the pan-genome is simply the sum of the genes in the reference genome and all the non-reference pan-genes.”

We note that a similar rationale for estimating pan-genome size and acknowledgement of limitations on determining the core genome was recognized during the construction of the pan-genome for Medicago (submitted about the same time as our manuscript), which I quote here (bold added): "Important caveats should be kept in mind when interpreting these results. Due to the incompleteness of the de novo Medicago assemblies (i.e., certain portions of genome were difficult to assemble), sequences present in one assembly but absent in others could have been due to technical artifact. This would have resulted in over-estimates of dispensable genome size. **By contrast, the pan-genome size estimate should be more robust since it surveys novel sequences across all accessions – and it is much less likely that a given genome region would be missed in all assemblies.**"⁸

In addition, as already addressed in the first revision, we demonstrated that no single line contributed a disproportionate number of genes to the high-confidence pan-genome: “To address the reviewer’s concern that a few lines may disproportionately inflate the pan-genome, we plotted the number of core, soft core, shell and cloud gene in each line (Fig. 2b). In addition, we plotted the number of high-quality non-reference genes whose transcripts did not map to the reference genome in Fig. S1d. There are no outliers on either graph indicating that our estimate of pan-genome size was not adversely affected by poor assembly quality of individual lines.”

Thus, the main conclusions from our work that the high-confidence pan-genome is much larger than the genome from any individual line and that shell genes make up nearly half of the pan-genome are robust. We do not make any firm conclusions about the size of the core genome because of the limitations of Illumina assemblies. To reiterate, uncertainty about the core/soft-core boundary does not alter the size of the pan-genome in relation to the genome of any individual line and by comparing the shell and core categories we are comparing genes that are really variable to genes that are core.

2. Controls and cross-validation

We have extensively validated our assemblies by several measures that go way beyond previous studies. This was described in my response to the initial reviews so I will just emphasize some points here. Assembly completeness as measured by assembly size and BUSCO scores both indicate that the vast majority of the genomes are highly complete. In fact, our assemblies are much more complete than any other pan-genome study and are on par with some reference genomes as described in the initial

response: “The average completeness of all assemblies is 95.2% as estimated by BUSCO, which makes our average assembly nearly as complete as the rice reference genome (*Oryza sativa*, MSU v7.0: 95.6%). Only five of our assemblies have a completeness score less than BUSCO 90% (supp. table 2). All of our assemblies have far greater completeness as compared to the recently published reference genome for *Oropetium thomaeum* (70.2%), which is a small low-repeat grass genome similar to *B. distachyon*²³.” We also included an assembly control where we assembled short reads sequences from the reference line and included that in our analyses. As expected, that assembly was extremely similar to the reference genome indicating that we did not have any large systematic errors in assembly or annotation. I am not aware of any other pan-genome study that included such a control.

To confirm the PAV variation detected in the assemblies we mapped raw reads from different lines and from the assembled line in question onto each assembly. By looking for areas with low read mapping we easily identified regions that are missing from one line. I pasted images for Fig. 1e and f below to show examples of what this looks like. In my initial response to reviewers’ comments I described in detail how we used read mapping on scales ranging from pairwise comparisons and single genes to entire population analysis. I won’t rehash that here, but I do want to emphasize that this is much more sensitive and accurate than picking a handful of genes for validation by PCR. It is not affected by the vagaries of PCR which is further limited by the fact that PCR is really only interrogating the primer sequence while read mapping looks at the entire variable sequence. Furthermore, since we sequenced so many lines we can place the pan-genome into a true population context for the first time. This is one of the main strengths of our work and is very interesting for a variety of biological reasons, but here I will just focus on validation. When we clustered the lines according to presence/absence or copy number variation across the ENTIRE genomes of all 54 lines we recovered the same population groups and relationships between lines. There is simply no way that we would have reconstructed those relationships if we had large errors in our PAV/CNV detection. This was described in my first response: “To extend this type of analysis across the entire pan-genome, in figure 2d we mapped reads from each individual line to the CDS from all pan-genes (each pan-gene cluster is represented by a single gene from the pan-gene cluster). The percentage of each gene covered by short reads is represented by color-coding. As is evident from the picture, the core genes are covered essentially 100% in all lines and the shell genes lack coverage from many lines as predicted from our gene-based pan-genome. When the lines are clustered using the read coverage percentage, the tree is essentially identical to the SNP-based phylogeny in figure 4. Furthermore, supplemental figure 4 shows a similar clustering based on pan-gene CNV which, again, produces essentially the same tree produced by the SNP-based analysis. This indicates that the observed PAV and CNV is consistent with the underlying biology and not due to random assembly artefacts.”

Taken together, this multi-faceted validation is consistent with the conclusions presented. The use of population genetics is a powerful new way to validate pan-genomes. Simply put, if our results were artefactual or if individual lines were aberrant then we would not have repeatedly recovered the same population structure by every analysis.

Figure 1e and f. Read mapping to validate PAV. (e) Raw reads, red and blue dashes, from 10 lines mapped onto the same genome assembly. Note that the lack of read support in 4 lines supports the PAV detected in the genome assemblies for this lines. (f) Read mapping can validate much larger variants than PCR as shown for the 400 kb interval here. Importantly, read mapping interrogates the entire sequence in question, not just primer sequences.

Finally, for further validation we compared our Illumina assembly for Bd21-3 to a new reference-quality PacBio assembly that is available as early release still under embargo on Phytozome. This assembly is of extremely high quality. (full assembly statistics at https://phytozome.jgi.doe.gov/pz/portal.html#!info?alias=Org_BdistachyonBd21_3_er) Ninety percent of the 1,100 Bd21-3 shell genes that were not in the *B. distachyon* Bd21 reference genome annotation aligned to the Bd21-3 PacBio assembly at greater than 99% identify (nucleotide level) over greater than 90 percent of the query sequence. Thus, by an independent measure one of our mid-quality assemblies is overwhelmingly accurate. We also observed that 77% of the 277 Bd21-3 cloud genes that were not in the *B. distachyon* Bd21 gene set aligned to the Bd21-3 PacBio assembly at greater than 99% identify (nucleotide level) over greater than 90 percent of the query sequence. This indicates that while the cloud genes are still largely accurate, the error rate is in fact higher than for the shell genes. This supports our rationale for breaking the pan-genome into four compartments. We hope this comparison of the novel genes across an ENTIRE genome for one of our mid-quality assemblies to a completely independent extremely high-quality reference assembly alleviates concerns about the quality of our assemblies. However, we did not add this analysis to the manuscript because the PacBio assembly is part of a different project with different collaborators and we don't have freedom to make that part of this project.

Consistency with previous studies.

As I mentioned previously, our results do not indicate that “that there are very few genes found in all lines” and our results are consistent with previous studies. For the reference line 44% of the high-confidence pan-genes are shared by all lines and 67% of pan-genes are shared by nearly all lines (core plus soft-core). This is consistent with several previous studies and the following text was added to the

discussion to indicate this: “While the *B. distachyon* pan-genome is larger than the pan-genomes reported for several other species, it is not completely unexpected based on results from other species. For example, an estimation of the maize pan-genome based on the pan-transcriptome¹² was of similar magnitude to the *B. distachyon* pan-genome. In addition, a study of 10 reference-guided assemblies of *Brassica oleracea* found that 20% of pan-genes showed PAV⁵ which is consistent with our simulation of the pan-genome size for 10 *B. distachyon* lines (Supp. Fig. 2A). Similarly, true-de-novo assemblies of 15 *Medicago* genomes found that 42% of genomic sequence was found in only some accessions⁸.” There are other studies that have described smaller pan-genomes in other species and this may be due to a combination of technical and biological reasons including:

We included more lines studies

We had more complete true *de novo* assemblies than any previous study

We selected our lines to maximize diversity

B. distachyon is a wild plant that has not undergone a domestication bottleneck

In addition, on a pairwise comparison any two *B. distachyon* lines share on average 90% of their genes which is actually higher than some pairwise comparisons of other plants. We added the following text to emphasize this “Despite the large overall size of the *B. distachyon* pan-genome, individuals in our study share the vast majority (90% on average) of their genes at the pairwise-level. Previous studies have observed greater amounts of pairwise PAV between individuals in other plant species^{15,16,18}. Thus, the amount of PAV that we observe in the compact genome of *B. distachyon* is likely to be dwarfed by PAV in larger, more complex genomes such as Maize¹⁵.” Significantly, all of our assemblies and other data is easily accessible through our website (we have been unable to get the assemblies and raw data for some other studies) so it can be easily used by any group.

The terminology of core, soft-core, cloud etc. was also questioned by another reviewer. While this may have been used once in a microbial pangenome, I recommend that the authors read the numerous plant pangenome papers where they will see that the two level nomenclature is used in all of them. The authors on one hand claim to have excellent assemblies but then introduce classifications and nomenclature which highlights the uncertainty in the results.

I agree with the reviewer that for most of the studies published to data that look at small number of genomes a binary classification system is adequate. However, since even the best assemblies and annotations will have errors and differences due to random factors I think it is very useful for studies that include large numbers of genomes (such as ours) to use categories that capture some of this uncertainty. I have already described above and in the manuscript why this is important. In any case, all the genes are included in our data release and available to the public through our BrachyPan website. Since this is such a new field there is no accepted nomenclature and, as I mentioned previously, another reviewer wanted us to add additional categories. The binary terminology of core versus dispensable has also been questioned by others (reference 17 below) in the light of whether it is misleading. For example, “dispensable” terminology implies that it is not necessary for survival, but in fact such sequences may be conditionally required for some populations to inhabit different environments.

Regarding the collapse of repetitive genes, the argument that some genes have assembled does not mean that all have. Even the best assemblies collapse some repetitive genes such as NRS LRRs, the important aspect is understanding how much collapsing there is and accounting for this in the interpretation of the results.

As mentioned above, our primary goal was to estimate the lower bound for the size of the *B. distachyon* pan-genome. We acknowledge that repetitive genes are more likely to be collapsed than other genes and we do not claim that the assemblies are perfect. Furthermore, since we intentionally collapsed similar genes into a single pan-gene cluster the collapsing of similar repeated genes during assembly will not significantly affect the size of the pan-genome. That said, our assemblies did capture clusters of NBS-LRR, leucine rich repeat receptor-like protein kinase and NB-ARC genes, all often found in clusters, remarkably well. In our initial revision we added a figure that addresses this concern. By comparing the NBS-LRR genes in our reference genome control Illumina assembly to a manually curated set of 119 NBS-LRR genes in the reference genome we showed that the majority of the clusters contained similar numbers of genes. In addition, the fact that all the assemblies and the reference genome contained similar numbers of three types of genes often found in multi-copy clusters (Supp. fig. f and g) indicates that there was no systematic bias in the assemblies and annotations.

Supp. Fig. 1e-g. Gene cluster analysis. (e) 119 pan-genome clusters associated with reference genes previously identified as NBS-LRRs are plotted in rows with the number of sequences within each individual colored according to the scale below the plot. Note how similar the reference assembly control is to the reference genome. (f) Number of genes annotated as PTHR24420 (Leucine rich repeat receptor-like protein kinase) and g, number of PF00931 NB-ARC protein, per inbred line.

Reviewer #2 (Remarks to the Author):

The authors have satisfactorily responded to the extensive comments from the earlier submission. The paper overall is sound as far as I can judge, and the findings are important and sufficiently well-documented.

A couple points the authors may choose to address in the final version:

1. In the pgph starting line 315, is what the authors refer to as Figure 4 in the text actually Fig 5? (not sure as the main figures were not included in the revised MS)

The text is correct. I think the reviewer was looking at the old figures.

2. The relationship of the two ASR phylogenies shown in Supp Fig 5a is not clear to me from the legend.

The tree on the right was removed.

Reviewer #3 (Remarks to the Author):

The current version of the manuscript has been widely improved and is of great interest.

Thus I agree for its publication

Reviewer #4 (Remarks to the Author):

Almost all of my concerns and criticism has been addressed and those that have been not addressed by the authors are either very minor or the authors gave a reasonable response on why they prefer to not address the issue.

I'd like to see the work and analysis published asap.

Klaus Mayer

- 1 Li, R. *et al.* Building the sequence map of the human pan-genome. *Nat Biotechnol* **28**, 57-63, doi:10.1038/nbt.1596 (2010).
- 2 Ossowski, S. *et al.* Sequencing of natural strains of *Arabidopsis thaliana* with short reads. *Genome Research* **18**, 2024-2033 (2008).
- 3 Gore, M. A. *et al.* A first-generation haplotype map of maize. *Science* **326**, 1115-1117, doi:10.1126/science.1177837 (2009).
- 4 Montenegro, J. D. *et al.* The pangenome of hexaploid bread wheat. *Plant J* **90**, 1007-1013, doi:10.1111/tpj.13515 (2017).
- 5 Golicz, A. A. *et al.* The pangenome of an agronomically important crop plant *Brassica oleracea*. *Nat Commun* **7**, 13390, doi:10.1038/ncomms13390 (2016).
- 6 Gan, X. *et al.* Multiple reference genomes and transcriptomes for *Arabidopsis thaliana*. *Nature* **477**, 419-423, doi:10.1038/nature10414 (2011).
- 7 Cao, J. *et al.* Whole-genome sequencing of multiple *Arabidopsis thaliana* populations. *Nature Genetics* **43**, 956-965, doi:10.1038/ng.911 (2011).
- 8 Zhou, P. *et al.* Exploring structural variation and gene family architecture with De Novo assemblies of 15 *Medicago* genomes. *BMC Genomics* **18**, 261, doi:10.1186/s12864-017-3654-1 (2017).
- 9 Li, Y. H. *et al.* De novo assembly of soybean wild relatives for pan-genome analysis of diversity and agronomic traits. *Nat Biotechnol* **32**, 1045-1052, doi:10.1038/nbt.2979 (2014).
- 10 Yao, W. *et al.* Exploring the rice dispensable genome using a metagenome-like assembly strategy. *Genome Biol* **16**, 187, doi:10.1186/s13059-015-0757-3 (2015).

- 11 Hirsch, C. N. *et al.* Insights into the maize pan-genome and pan-transcriptome. *Plant Cell* **26**, 121-135, doi:10.1105/tpc.113.119982 (2014).
- 12 Jin, M. L. *et al.* Maize pan-transcriptome provides novel insights into genome complexity and quantitative trait variation. *Scientific Reports* **6**, doi:Artn 18936
10.1038/Srep18936 (2016).
- 13 Lai, J. *et al.* Genome-wide patterns of genetic variation among elite maize inbred lines. *Nat Genet* **42**, 1027-1030, doi:10.1038/ng.684 (2010).
- 14 Hardigan, M. A. *et al.* Genome Reduction Uncovers a Large Dispensable Genome and Adaptive Role for Copy Number Variation in Asexually Propagated *Solanum tuberosum*. *Plant Cell* **28**, 388-405, doi:10.1105/tpc.15.00538 (2016).
- 15 Jiao, Y. *et al.* Improved maize reference genome with single-molecule technologies. *Nature* **546**, 524-527, doi:10.1038/nature22971 (2017).
- 16 Springer, N. M. *et al.* Maize inbreds exhibit high levels of copy number variation (CNV) and presence/absence variation (PAV) in genome content. *PLoS Genet* **5**, e1000734, doi:10.1371/journal.pgen.1000734 (2009).
- 17 Marroni, F., Pinosio, S. & Morgante, M. Structural variation and genome complexity: is dispensable really dispensable? *Curr Opin Plant Biol* **18**, 31-36, doi:10.1016/j.pbi.2014.01.003 (2014).
- 18 Flagel, L. E., Willis, J. H. & Vision, T. J. The standing pool of genomic structural variation in a natural population of *Mimulus guttatus*. *Genome Biol Evol* **6**, 53-64, doi:10.1093/gbe/evt199 (2014).
- 19 Freeling, M. *et al.* Many or most genes in *Arabidopsis* transposed after the origin of the order Brassicales. *Genome Res* **18**, 1924-1937, doi:10.1101/gr.081026.108 (2008).
- 20 Choulet, F. *et al.* Megabase level sequencing reveals contrasted organization and evolution patterns of the wheat gene and transposable element spaces. *Plant Cell* **22**, 1686-1701 (2010).
- 21 Woodhouse, M. R. *et al.* Following tetraploidy in maize, a short deletion mechanism removed genes preferentially from one of the two homologs. *PLoS Biol* **8**, e1000409, doi:10.1371/journal.pbio.1000409 (2010).
- 22 International Brachypodium, I. Genome sequencing and analysis of the model grass *Brachypodium distachyon*. *Nature* **463**, 763-768, doi:10.1038/nature08747 (2010).
- 23 VanBuren, R. *et al.* Single-molecule sequencing of the desiccation-tolerant grass *Oropetium thomaeum*. *Nature* **527**, 508-511, doi:10.1038/nature15714 (2015).

Reviewers' Comments:

Reviewer #1 (Remarks to the Author):

Many thanks for the opportunity to read the improved version of the manuscript. I understand that it can be frustrating having to update manuscripts due to other related work being published while under review, but I believe it is worth it to ensure that the final published version is current and most relevant. I do not believe it is necessary to paint other pangenome studies in such negative light, but at least the authors have acknowledged the other work in the field which is all that was required. The authors have made significant improvements with this version, and while I still consider the terminology may be confusing, by placing his in context in the manuscript they have made a sufficiently strong case for its use. Overall, the manuscript is very much improved and I have no further suggestions.

Reviewer #3 (Remarks to the Author):

As I stated before the previous revision was ok for me already.
The current one is even more excellent and I am waiting for this paper to be published soon

Francois Sabot